# META TEMPORAL POINT PROCESSES

**Wonho Bae**
University of British Columbia & Borealis AI
`whbae@cs.ubc.ca`

**Mohamed Osama Ahmed**
Borealis AI
`mohamed.o.ahmed@borealisai.com`

**Frederick Tung**
Borealis AI
`frederick.tung@borealisai.com`

**Gabriel L. Oliveira**
Borealis AI
`gabriel.oliveira@borealisai.com`

## ABSTRACT

A temporal point process (TPP) is a stochastic process where its realization is a sequence of discrete events in time. Recent work in TPPs model the process using a neural network in a supervised learning framework, where a training set is a collection of all the sequences. In this work, we propose to train TPPs in a meta learning framework, where each sequence is treated as a different task, via a novel framing of TPPs as neural processes (NPs). We introduce context sets to model TPPs as an instantiation of NPs. Motivated by attentive NP, we also introduce local history matching to help learn more informative features. We demonstrate the potential of the proposed method on popular public benchmark datasets and tasks, and compare with state-of-the-art TPP methods.

## 1 INTRODUCTION

With the advancement of deep learning, there has been growing interest in modeling temporal point processes (TPPs) using neural networks. Although the community has developed many innovations in how neural TPPs encode the history of past events (Biloš et al., 2021) or how they decode these representations into predictions of the next event (Shchur et al., 2020; Lin et al., 2022), the general training framework for TPPs has been *supervised learning* where a model is trained on a collection of all the available sequences. However, supervised learning is susceptible to overfitting, especially in high noise environments, and generalization to new tasks can be challenging.

In recent years, *meta learning* has emerged as an alternative to supervised learning as it aims to adapt or generalize well on new tasks, which resembles how humans can learn new skills from a few examples. Inspired by this, we propose to train TPPs in a meta learning framework. To this end, we treat each sequence as a "task", since it is a realization of a stochastic process with its own characteristics. For instance, consider the pickup times of taxis in a city. The dynamics of these event sequences are governed by many factors such as location, weather and the routine of a taxi driver, which implies the pattern of each sequence can be significantly different from each other. Under the supervised learning framework, a trained model tends to capture the patterns seen in training sequences well, but it easily breaks on unseen patterns.

As the goal of modeling TPPs is to estimate the true probability distribution of the next event time given the previous event times, we employ Neural Processes (NPs), a family of the model-based meta learning with stochasticity, to explain TPPs. In this work, we formulate neural TPPs as NPs by satisfying some conditions of NPs, and term it as Meta TPP. Inspired by the literature in NP, we further propose the Meta TPP with a cross-attention module, which we refer to as Attentive TPP. We demonstrate the strong potential of the proposed method through extensive experiments.

Our contributions can be summarized as follows,

- To the best of our knowledge, this is the first work that formulates the TPP problem in a meta learning framework, opening up a new research direction in neural TPPs.
- Inspired by the NP literature, we present a conditional meta TPP formulation, followed by a latent path extension, culminating with our proposed Attentive TPP model.

- The experimental results show that our proposed Attentive TPP model achieves state-of-the-art results on four widely used TPP benchmark datasets, and is more successful in capturing periodic patterns on three additional datasets compared to previous methods.

- We demonstrate that our meta learning TPP approach can be more robust in practical deployment scenarios such as noisy sequences and distribution drift.

## 2 PRELIMINARIES

**Neural processes.** A general form of optimization objective in **supervised learning** is defined as,

$$\theta^* = \arg\max_{\theta} \mathbb{E}_{B \sim p(\mathcal{D})} \left[ \sum_{(x,y) \in B} \log p_\theta(y \mid x) \right] \tag{1}$$

where $\mathcal{D} := \{(x^{(i)}, y^{(i)})\}_{i=1}^{|\mathcal{D}|}$ for an input $x$ and label $y$, and $B$ denotes a mini-batch set of $(x, y)$ data pairs. Here, the goal is to learn a model $f$ parameterized by $\theta$ that maps $x$ to $y$ as $f_\theta : x \to y$.

In recent years, **meta learning** has emerged as an alternative to supervised learning as it aims to adapt or generalize well on new tasks (Santoro et al., 2016), which resembles how humans learn new skills from few examples. In meta learning, we define a meta dataset, a set of different tasks, as $\mathcal{M} := \{\mathcal{D}^{(i)}\}_{i=1}^{|\mathcal{M}|}$. Here, $\mathcal{D}^{(i)}$ is a dataset of $i$-th task consisting of a context and target set as $\mathcal{D} := \mathcal{C} \cup \mathcal{T}$. The objective of meta learning is then defined as,

$$\theta^* = \arg\max_{\theta} \mathbb{E}_{B_\mathcal{D} \sim p(\mathcal{M})} \left[ \sum_{(\mathcal{C},\mathcal{T}) \in B_\mathcal{D}} \log p_\theta(\mathcal{Y}_\mathcal{T} \mid \mathcal{X}_\mathcal{T}, \mathcal{C}) \right] \tag{2}$$

where $B_\mathcal{D}$ denotes a mini-batch set of tasks. Also, $\mathcal{X}_\mathcal{T}$ and $\mathcal{Y}_\mathcal{T}$ represent inputs and labels of a target set, respectively. Unlike supervised learning, the goal is to learn a mapping from $x$ to $y$ given $\mathcal{C}$: more formally, $f_\theta(\cdot, \mathcal{C}) : x \to y$. Although meta learning is a powerful framework to learn fast adaption to new tasks, it does not provide uncertainty for its predictions, which is becoming more important in modern machine learning literature as a metric to measure the reliability of a model.

To take the uncertainty into account for meta learning, **Neural processes** (NPs) have been proposed (Garnelo et al., 2018b;a). Instead of finding point estimators as done in regular meta learning models, NPs learn a probability distribution of a label $y$ given an input $x$ and context set $\mathcal{C}$: $p_\theta(y|x, \mathcal{C})$. In this work, we frame TPPs as meta learning instead of supervised learning, for the first time. To this end, we employ NPs to incorporate the stochastic nature of TPPs. In Section 3.1, we propose a simple modification of TPPs to connect them to NPs, which enables us to employ a rich line of works in NPs to TPPs as described in Section 3.2 and Section 3.3.

**Neural temporal point processes.** TPPs are stochastic processes where their realizations are sequences of discrete events in time. In notations, a collection of event time sequences is defined as $\mathcal{D} := \{s^{(i)}\}_{i=1}^{|\mathcal{D}|}$ where $s^{(i)} = (\tau_1^{(i)}, \tau_2^{(i)}, \ldots, \tau_{L_i}^{(i)})$ and $L_i$ denotes the length of $i$-th sequence. The history of studying TPPs started decades ago (Daley & Vere-Jones, 2003), but in this work, we focus on neural TPPs where TPPs are modeled using neural networks (Shchur et al., 2021). As described in Figure 1a, a general form of neural TPPs consists of an encoder, which takes a sequence of previous event times and outputs a history embedding, and a decoder which takes the history embedding and outputs probability distribution of the time when the next event happens.

Previous works of neural TPPs are auto-regressively modeled in a supervised learning framework. More formally, the objective of neural TPPs are defined as,

$$\theta^* = \arg\max_{\theta} \mathbb{E}_{B \sim p(\mathcal{D})} \left[ \sum_{i=l}^{|B|} \sum_{l=1}^{L_i-1} \log p_\theta(\tau_{l+1}^{(i)} \mid \tau_{\leq l}^{(i)}) \right] \tag{3}$$

where $B \sim p(\mathcal{D})$ denotes a mini-batch of event time sequences. To frame TPPs as NPs, we need to define a target input and context set shown in Equation (2), from an event time history $\tau_{\leq l}$, which will be described in the following section.

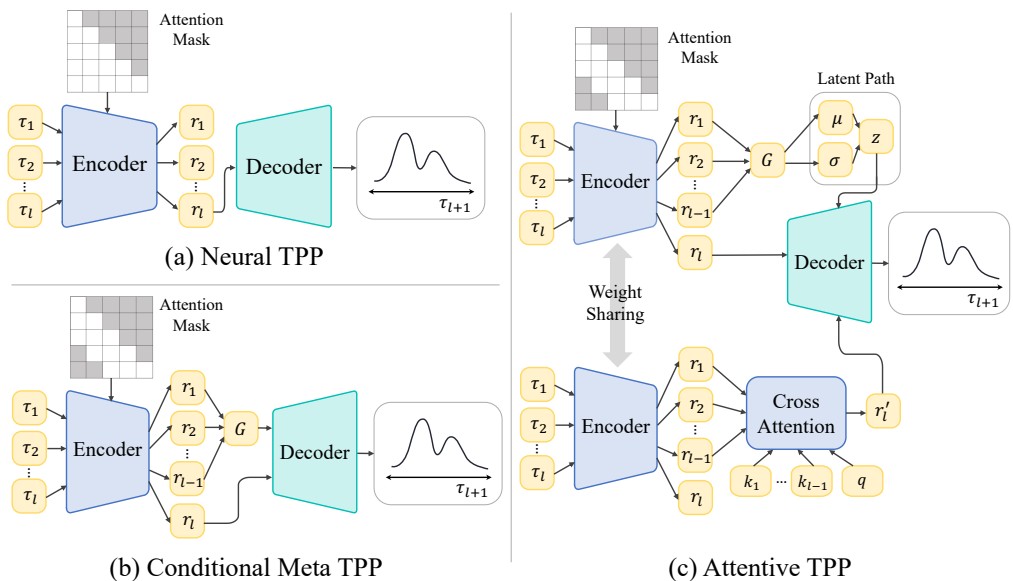

Figure 1: Overall architectures of TPP models.

# 3 META TEMPORAL POINT PROCESS AND ITS VARIANTS

## 3.1 TEMPORAL POINT PROCESSES AS NEURAL PROCESSES

To frame TPPs as NPs, we treat each event time sequence $s$ as a task for meta learning, which intuitively makes sense since each sequence is a realization of a stochastic process. For instance, the transaction times of different account holders are very different from each other due to many factors including an account holder's financial status and characteristics.

With the new definition of tasks, we define a target input and context set for a conditional probability distribution of meta learning shown in Equation (2), using previous event times $\tau_{\leq l}$. There are many ways to define them but a target input and context set need to be semantically aligned since the target input will be an element of the context set for the next event time prediction. Hence, we define a target input for $\tau_{l+1}$ as the latest "local history" $\tau_{l-k+1:l}$ where $k$ is the window size of the local history. Similarly, a context set for $\tau_{l+1}$ is defined as $\mathcal{C}_l := \{\tau_{t-k+1:t}\}_{t=1}^{l-1}$. Here, if $t - k \leq 0$, we include event times from $\tau_1$. With Transformer structure, it is easy to efficiently compute the feature embeddings for the context set $\mathcal{C}$. Figure 1b shows a schematic of the Conditional Meta TPP with a mask (shaded) used for an example case of 5 event times with a local history window size of $k = 3$. Then, the feature embedding $r_l$ contains information of $\tau_{l-k+1:l}$. With the notations for target inputs and context sets, we propose the objective of TPPs in a meta learning framework as,

$$\theta^* = \arg\max_\theta \mathbb{E}_{B \sim p(\mathcal{D})} \left[ \sum_{i=l}^{|B|} \sum_{l=1}^{L_i-1} \log p_\theta(\tau_{l+1}^{(i)} \mid \tau_{l-k+1:l}^{(i)}, \mathcal{C}_l^{(i)}) \right]. \tag{4}$$

Note that we have only one target label $\tau_{l+1}^{(i)}$ to predict per event unlike the general meta learning objective in Equation (2) where usually $|\mathcal{T}| > 1$. It is because TPP models in general are trained to predict the next event time. Modeling TPPs to predict multiple future event times would be an interesting future work, but it is out of scope of this work.

**Requirements for neural processes.** Let $\mathcal{X}_\mathcal{T} := \{x_i\}_{i=1}^{|\mathcal{T}|}$ and $\mathcal{Y}_\mathcal{T} := \{y_i\}_{i=1}^{|\mathcal{T}|}$ be a set of target inputs and labels, respectively, and $\pi$ be an arbitrary permutation of a set. To design NP models, it is required to satisfy the following two conditions.

**Condition 3.1** (Consistency over a target set). *A probability distribution $p_\theta$ is consistent if it is consistent under permutation: $p_\theta(\mathcal{Y}_\mathcal{T} \mid \mathcal{X}_\mathcal{T}, \mathcal{C}) = p_\theta(\pi(\mathcal{Y}_\mathcal{T}) \mid \pi(\mathcal{X}_\mathcal{T}), \mathcal{C})$, and marginalization: $p_\theta(y_{1:m} \mid \mathcal{X}_\mathcal{T}, \mathcal{C}) = \int p_\theta(y_{1:n} \mid \mathcal{X}_\mathcal{T}, \mathcal{C}) \, dy_{m+1:n}$ for any positive integer $m < n$.*

**Condition 3.2** (Permutation invariance over a context set). $p_\theta(\mathcal{Y}_\mathcal{T} \,|\, \mathcal{X}_\mathcal{T}, \mathcal{C}) = p_\theta(\mathcal{Y}_\mathcal{T} \,|\, \mathcal{X}_\mathcal{T}, \pi(\mathcal{C}))$

According to Kolmogorov extension theorem (Oksendal, 2013), a collection of finite-dimensional distributions is defined as a stochastic process if condition 3.1 is satisfied. In NP literature, condition 3.1 is satisfied through factorization: it assumes target labels are independent to each other given a target input and a context set $\mathcal{C}$, in other words, $p_\theta(\mathcal{Y}_\mathcal{T} \,|\, \mathcal{X}_\mathcal{T}, \mathcal{C}) = \Pi_{i=1}^{|\mathcal{T}|} p_\theta(y_i \,|\, x_i, x_{<i}, y_{<i}, \mathcal{C}) \approx \Pi_{i=1}^{|\mathcal{T}|} p_\theta(y_i \,|\, x_i, \mathcal{C})$ (Dubois et al., 2020). This assumption can be unrealistic if target labels are strongly dependent to previous target inputs and labels even after context representations are observed. It is, however, not necessary to assume factorization to make TPPs as NPs. As previously mentioned, we only care about predicting the next event time, which means $|\mathcal{Y}_\mathcal{T}| = 1$. When a set contains only one element, its permutation is always itself. More formally, the consistency under permutation of Condition 3.1 in TPPs: $p_\theta(\tau_{l+1} \,|\, \tau_{l-k+1:l}, \mathcal{C}_l) = p_\theta(\pi(\tau_{l+1}) \,|\, \pi(\tau_{l-k+1:l}), \mathcal{C}_l)$, is satisfied since $\pi(\{\tau_{l+1}\}) = \{\tau_{l+1}\}$ and $\pi(\{\tau_{l-k+1:l}\}) = \{\tau_{l-k+1:l}\}$. Also, the marginalization under permutation in Condition 3.1 is satisfied as marginalization is not applicable for $p_\theta(\tau_{l+1} \,|\, \tau_{l-k+1:l}, \mathcal{C}_l)$ since the target label set $\tau_{l+1}$ contains only one element.

Recall that NP models learn a probability distribution of a target label $p_\theta$ given a target input and context set. For computational efficiency (to make inference $\mathcal{O}(|\mathcal{C}| + |\mathcal{T}|)$ time), the feature representation of $\mathcal{C}$ should be invariant to the size of the context set, for which Condition 3.2 is required. To satisfy Condition 3.2, we average-pool all the context features $r_1, r_2, \ldots r_{l-1}$ to generate the global feature for a task $G$ as shown in Figure 1b, and term it as *conditional Meta TPP* following the terminology used in the NP literature. Each context feature $r_1, r_2, \cdots, r_{l-1}$ represents a feature from a transformer encoder such as Transformer Hawkes Processes (THP), that encodes the corresponding local history of the context set $\mathcal{C}_l$. For instance, $r_i$ contains information of $\tau_{i-k+1:i}$. To make $r_i$ only encode the subset of previous event times $\tau_{i-k+1:i}$ (instead of the whole previous event times $\tau_{\leq i}$), we mask out events that are outside of the local history window using an attention mask as shown in Figure 1(b) and (c), which is different from a regular attention mask shown in Figure 1(a). Using the permutation invariant feature $G$ not only satisfies Condition 3.2, but also lets the decoder approximate the probability distribution of a target label given both a target input and context set instead of just a target input. Now that we satisfy both requirements with a new architectural design, we can treat TPPs as NPs.

**Implementation.** It can be expensive to compute the individual context feature $r_t$ for all $1 \leq t < l$, from each element of the context set $\tau_{t-k+1:t} \in \mathcal{C}_l$: the time complexity of computing all the context features for a sequence is $\mathcal{O}(L^2)$. Instead of passing each element of a context set, using the Transformer architecture (Vaswani et al., 2017), we can simply pass the event times to obtain all the context features at once, of which time complexity is $\mathcal{O}(kL)$ where $k$ is the window size of a local history. To this end, we employ the THP as the encoder. Please refer to Zuo et al. (2020) for details.

## 3.2 Meta Temporal Point Process

In the NP literature, NPs are generally modeled as latent variable models. Instead of using the deterministic global feature $G$ as an input to the decoder (Garnelo et al., 2018a), a latent variable $z$ is sampled from a probability distribution *e.g.* multi-variate Gaussian, using parameters inferred from the global feature $G$ (Garnelo et al., 2018b). As it is intractable to compute the log-likelihood for a latent variable model, *amortized variational inference* (VI) can be used to approximate inference. In the setting of TPPs, the evidence lower bound (ELBO) of variational inference with an inference network $p_\theta(z \,|\, \mathcal{C}_L)$ can be derived as,

$$\log p_\theta(\tau_{l+1} \,|\, \tau_{l-k+1:l}, \mathcal{C}_l) = \log \int p_\theta(\tau_l \,|\, \tau_{l-k+1:l}, z) p_\theta(z \,|\, \mathcal{C}_l) dz \tag{5}$$

$$\geq \mathbb{E}_z \left[ \log p_\theta(\tau_{l+1} \,|\, \tau_{l-k+1:l}, z) \right] - KL(p_\theta(z \,|\, \mathcal{C}_L) \,|\, p_\theta(z \,|\, \mathcal{C}_l)) \tag{6}$$

$$\approx \frac{1}{N} \sum_{n=1}^{N} \log p_\theta(\tau_l \,|\, \tau_{l-k+1:l}, z_n) - KL(p_\theta(z \,|\, \mathcal{C}_L) \,|\, p_\theta(z \,|\, \mathcal{C}_l)) \tag{7}$$

where $N$ denotes the number of samples of $z \sim p_\theta(z \,|\, \mathcal{C}_L)$ for Monte-Carlo approximation. Here, $p_\theta(z \,|\, \mathcal{C}_L)$ is the posterior given the context at the last ($L$-th) event, which contains all the events of the sequence $s$ (it is accessible in training time). Minimizing $KL$-divergence between $p_\theta(z \,|\, \mathcal{C}_L)$ and $p_\theta(z \,|\, \mathcal{C}_l)$ is to make the global latent variable $z$ inferred from $\mathcal{C}_l$ to be similar to the latent

variable of a sequence $z$ from $\mathcal{C}_L$, in training time. To sample $z$, we use the reparameterization trick as $z = \mu + \sigma \odot \epsilon$ where $\epsilon \sim \mathcal{N}(0, I)$ as described in the latent path of Figure 1c. In inference, we approximate evaluation metrics such as negative log-likelihood or root mean squared error using Monte-Carlo samples. But, as we do not have access to $\mathcal{C}_L$ at $l$-th event when $l < L$, we use $z$ from $p_\theta(z \,|\, \mathcal{C}_l)$. The detailed description of the evaluation metrics are provided in Appendix C.

An advantage of a latent variable model is that it captures stochasticity of functions, which can be particularly beneficial to model TPPs since TPPs are stochastic processes. Experiments in Section 5.4 demonstrate it indeed helps to model TPPs over the deterministic case. In particular, it is robust to noises (Section 5.2). We term the latent variable model as *Meta TPP* throughout the paper.

**Discussion.** The existing TPP models treat all event sequences as realization of the same process whereas the Meta TPP treats each sequence as a realization of a distinct stochastic process. We achieve this by conditioning on the global latent feature $z$ that captures task-specific characteristics. For $z$ to be task-specific, it has to be distinct for different sequences but similar throughout different events $l \in [1, L-1]$ within the same sequence. It is natural for the global features to be distinct by sequence but we need further guidance to make the global feature shared across all the event times in a sequence. Due to the permutation invariance constraint implemented in average-pooling, $z$ cannot be very different at different event time: adding some addition context feature $r_i$ will not change $G$ as well as $z$ much. In addition, the KL-divergence between $p_\theta(z \,|\, \mathcal{C}_L)$ and $p_\theta(z \,|\, \mathcal{C}_l)$ further enhances the task-specific characteristics of $z$. We provide more detailed discussion in Appendix B

### 3.3 ATTENTIVE TEMPORAL POINT PROCESS

Early works in NPs suffered from the underfitting problem. To alleviate this, Kim et al. (2019) proposed AttentiveNP, which explicitly attends the elements in a context set to obtain a better feature for target inputs. Inspired by this, we add a cross-attention module that considers the similarity between the feature of a target input and previous event times as described in Figure 1c. Given the local history (context) features $r_1, r_2, \ldots r_{l-1}$ at $l$-th time step, the key-query-value pairs $K \in \mathbb{R}^{l-1 \times D}, q \in \mathbb{R}^{1 \times D}$, and $V \in \mathbb{R}^{l-1 \times D}$ for the cross-attention, are computed using their corresponding projection weights $W_K \in \mathbb{R}^{D \times D}$, $W_Q \in \mathbb{R}^{D \times D}$ as,

$$K = R \cdot W_K, \; q = r_l^{\,T} \cdot W_Q, \; V = R \text{ where } R = [r_1, r_2, \ldots, r_{l-1}]^T. \tag{8}$$

Here, K corresponds to $[k_1, k_2, \ldots, k_{l-1}]^T$ in Figure 1c. The feature of $i$-th attention head $h_i$ are then computed as follows,

$$h_i = Softmax(q \cdot K^T / \sqrt{D}) \cdot V. \tag{9}$$

With $W \in \mathbb{R}^{HD \times D}$ and some fully connected layers denoted as $FC$, $r_l' \in \mathbb{R}^{1 \times D}$ is computed as,

$$r_l' = FC([h_1, h_2, \ldots, h_H] \cdot W). \tag{10}$$

Finally, the decoder takes the concatenated feature of $z, r_l$, and $r_l'$ as an input to infer a distribution.

In the TPP setting, it is common that there are multiple periodic patterns in the underlying stochastic process. The cross-attention module provides an inductive bias to a model that the repeating event subsequences should have similar features. Our experiments in Section 5.2 demonstrate that the explicit attention helps to model TPPs in general, especially when there are periodic patterns.

**Decoder.** The decoder takes the concatenated feature of the global latent feature $z$, target input feature $r_l$ that encodes $\tau_{l-k:l-1}$, and attention feature $r'$ from the attention module. For the Meta TPP (without the attention module), the decoder takes as input the concatenated feature of $z$ and $r_l$. Here, $z, r_l$, and $r'$ are all $D$-dimensional vectors. The decoder consists of two fully connected layers, and the input and hidden dimension of the decoder layers are either $2D$ or $3D$ depending on whether we use the feature from the attention module $r'$.

The decoder outputs the parameters of the probability distribution of the next event time or $p_\theta(\tau_{l+1} \,|\, \tau_{l-k+1:l}, z_m)$. Inspired by the intensity-free TPP (Shchur et al., 2020), we use a mixture of log-normal distributions to model the probability distribution. Formally, for $l \in [1, L-1]$, $\tau_{l+1} \sim MixLogNorm(\boldsymbol{\mu}_{l+1}, \boldsymbol{\sigma}_{l+1}, \boldsymbol{\omega}_{l+1})$ where $\boldsymbol{\mu}_{l+1}$ are the mixture means, $\boldsymbol{\sigma}_{l+1}$ are the standard deviations, and $\boldsymbol{\omega}_{l+1}$ are the mixture weights.

## 4 RELATED WORK

**Neural temporal point processes.** Neural temporal point processes (NTPPs) have been proposed to capture complex dynamics of stochastic processes in time. They are derived from traditional temporal point processes (Hawkes, 1971; Isham & Westcott, 1979; Daley & Vere-Jones, 2003). Models based on RNNs are proposed by (Du et al., 2016) and (Mei & Eisner, 2017) to improve NTPPs by constructing continuous-time RNNs. More recent works use Transformers to capture long-term dependency (Kumar et al., 2019; Zhang et al., 2020; Zuo et al., 2020; Yang et al., 2022; Zhang et al., 2022). (Omi et al., 2019; Shchur et al., 2020; Sharma et al., 2021) propose intensity-free NTPPs to directly model the conditional distribution of event times.

Omi et al. (2019) propose to model a cumulative intensity with a neural network. But, it suffers from problems that the probability density is not normalised and negative event times receives non-zero probabilities. Alternatively, Shchur et al. (2020) suggest modelling conditional probability density by log-normal mixtures. Transformer-based models like Zuo et al. (2020); Zhang et al. (2020) propose to leverages the self-attention mechanism to capture long-term dependencies. Another class of TPP methods called Neural Flows (Biloš et al., 2021), are proposed to model temporal dynamics with ordinary differential equations learned by neural networks. Unlike the previous TPP methods, we frame TPPs as meta learning (not supervised learning) for the first time.

**Neural processes.** Meta learning is a learning framework that aims to adapt or generalize well on new tasks. There are three approaches in meta learning: metric-based (Koch et al., 2015; Vinyals et al., 2016; Sung et al., 2018; Snell et al., 2017), model-based (Santoro et al., 2016; Munkhdalai & Yu, 2017; Grant et al., 2018) and optimization-based (Finn et al., 2017; 2018; Nichol et al., 2018). Neural processes (NPs) is the model-based meta learning with stochasticity. Garnelo et al. (2018a) propose a conditional neural process as a new formulation to approximate a stochastic process using neural network architecture. It succeeds the advantage of Gaussian Processes (GPs) as it can estimate the uncertainty of its predictions, without having expensive inference time. Garnelo et al. (2018b) generalize a conditional neural process by adding latent variables, which are approximated using variational inference. Although NPs can adapt to new tasks quickly without requiring much computation, it suffers from underfitting problem. To alleviate it, Kim et al. (2019) propose a cross-attention module, which explicitly attends the elements in the context set to obtain better representations for the elements in the target set. As another way to address the underfitting problem, Gordon et al. (2020) propose a set convolutional layer under the assumption of translation equivariance of inputs and outputs, which is expanded to the latent variable counterpart in Foong et al. (2020).

Transformer NP (Nguyen & Grover, 2022) is the most relevant work to ours. Although it also models event sequences, it focuses on modeling regular time series: discrete and regularly-spaced time inputs with corresponding label values. TPPs are different as they are continuous and irregularly-spaced time sequences not necessarily with corresponding label values.

## 5 EXPERIMENTS

### 5.1 EXPERIMENT SETTING

**Datasets.** To compare the effectiveness of models, we conduct experiments on 4 popular benchmark datasets – Stack Overflow, Mooc, Reddit, and Wiki, and 3 datasets with strong periodic patterns we introduce – Sinusoidal wave, Uber, and NYC Taxi. Please refer to Appendix H for details.

**Metrics.** We use the root mean squared error (RMSE) as the main metric along with the negative log-likelihood (NLL) as a reference since NLL can go arbitrary low if probability density is placed mostly on the ground truth event time. RMSE may not be a good metric, either, if one ignores stochastic components of TPPs and directly trains a baseline on the ground truth event times to obtain point estimations of event times (Shchur et al., 2021). We train all the methods on NLL and obtain RMSE in test time to not abuse RMSE scores, keeping stochastic components of TPPs. For marked TPP datasets, we extend the proposed method to make class predictions, and report accuracy. For details about marked cases, please refer to Appendix G.

**Baselines.** We use intensity-free TPP (Shchur et al., 2020), Neural flow (Biloš et al., 2021), and Transformer Hawkes Processes (THP) (Zuo et al., 2020) as baselines. For intensity-free TPP and

| Methods | Stack Overflow | | | Mooc | | | Reddit | | | Wiki | | |
|---|---|---|---|---|---|---|---|---|---|---|---|---|
| | RMSE | NLL | Acc | RMSE | NLL | Acc | RMSE | NLL | Acc | RMSE | NLL | Acc |
| Intensity-free | 3.64 | 3.66 | 0.43 | 0.31 | 0.94 | **0.40** | 0.18 | 1.09 | 0.60 | 0.60 | 7.76 | **0.26** |
| | (0.26) | (0.02) | (0.005) | (0.006) | (0.03) | **(0.004)** | (0.006) | (0.04) | (0.008) | (0.05) | (0.40) | **(0.03)** |
| Neural flow | – | – | – | 0.47 | 0.43 | 0.30 | 0.32 | 1.30 | 0.60 | 0.56 | 11.55 | 0.05 |
| | – | – | – | (0.006) | (0.02) | (0.04) | (0.04) | (0.33) | (0.07) | (0.05) | (2.22) | (0.01) |
| THP$^+$ | 1.68 | 3.28 | **0.46** | 0.18 | 0.13 | 0.38 | 0.26 | 1.20 | 0.60 | 0.17 | **6.25** | 0.23 |
| | (0.16) | (0.02) | **(0.004)** | (0.005) | (0.02) | (0.004) | (0.005) | (0.04) | (0.007) | (0.02) | **(0.39)** | (0.03) |
| Attentive TPP | **1.15** | **2.64** | **0.46** | **0.16** | **-0.72** | 0.36 | **0.11** | **0.03** | **0.60** | **0.15** | **6.25** | 0.25 |
| | **(0.02)** | **(0.02)** | **(0.004)** | **(0.004)** | **(0.02)** | (0.003) | **(0.002)** | **(0.04)** | **(0.007)** | **(0.01)** | **(0.38)** | (0.03) |

Table 1: Comparison of the Attentive TPP to the state-of-the-art methods on a bootstrapped test sets.

| Methods | Sinusoidal | | Uber | | NYC Taxi | |
|---|---|---|---|---|---|---|
| | RMSE | NLL | RMSE | NLL | RMSE | NLL |
| Intensity-free | 1.29 (0.08) | 0.88 (0.02) | 51.23 (2.89) | 4.46 (0.02) | 46.59 (26.16) | 2.06 (0.07) |
| Neural flow | **1.13 (0.07)** | 0.99 (0.02) | – | – | – | – |
| THP$^+$ | 1.72 (0.10) | 0.84 (0.02) | 90.25 (4.53) | 3.63 (0.03) | 10.31 (0.47) | **2.00 (0.01)** |
| Attentive TPP (Ours) | 1.45 (0.11) | **0.66 (0.02)** | **22.11 (1.94)** | **2.89 (0.04)** | **8.92 (0.42)** | **2.00 (0.009)** |

Table 2: Experiment results on bootstrapped test sets with strong periodic patterns.

neural flow, we add the survival time of the last event to NLL and fix some bugs specified in their public repositories. THP and its variants are originally based on intensity: they predict intensities from which log-likelihood and expectation of event times are computed. It is, however, computationally expensive to compute them as it requires to compute integrals: especially, to compute the expected event times, it requires to compute double integrals, which can be quite expensive and complex to compute even with thinning algorithms described in Mei & Eisner (2017). To work around it without losing performance, we add the mixture of log-normal distribution proposed in (Shchur et al., 2020) as the decoder, and we call it THP$^+$. For fair comparison, we fix the number of parameters of the models in between 50K and 60K except the last fully-connected layer for class predictions since it depends on the number of classes.

**Hyperparameters.** We grid-search on every combination of dataset and method for learning rate $\in \{0.01, 0.001, 0.0001, 0.00001\}$ and weight decay $\in \{0.01, 0.001, 0.0001, 0.00001\}$ for fair comparison. We bootstrap for 200 times on test sets to obtain the mean and standard deviation (in parentheses) for the metrics in Figure 2a and Table 1–3 following Yang et al. (2022). All the other hyperparameters are fixed throughout the experiments, and are reported in Appendix I. Our implementation is publicly available at `https://github.com/BorealisAI/meta-tpp`.

## 5.2 EXPERIMENT RESULTS

In this section, we begin by comparing our attentive meta temporal point process (denoted as Attentive TPP) with state-of-the-art supervised TPP methods on 4 popular benchmarks. We then investigate how Attentive TPP captures periodic patterns, and show how Attentive TPP can be used to impute missing events in noisy sequences. Finally, we consider robustness under distribution drift.

**Comparison with state-of-the-art methods.** Table 1 summarizes our comparison of Attentive TPP with state-of-the-art baselines – intensity-free (Shchur et al., 2020), neural flow (Biloš et al., 2021)[1], and THP$^+$ (Zuo et al., 2020) on the Stack Overflow, Mooc, Reddit, and Wiki benchmarks. THP$^+$ generally performs better than the intensity-free and neural flow baselines. Attentive TPP further improves over THP$^+$ on all datasets and metrics except for mark accuracy on Mooc and Wiki.

---

[1]Neural flow results on Uber, NYC Taxi and Stack Overflow (in Table 1) datasets are missing because the official implementation runs into NaN values for long event sequences in inversion step.

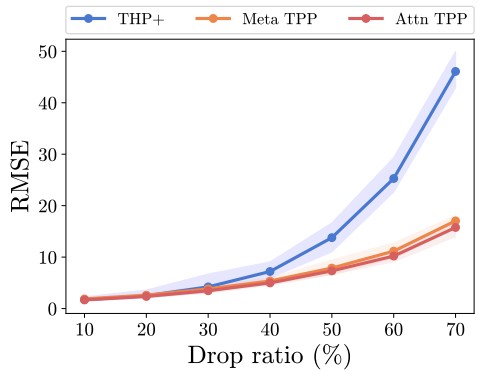

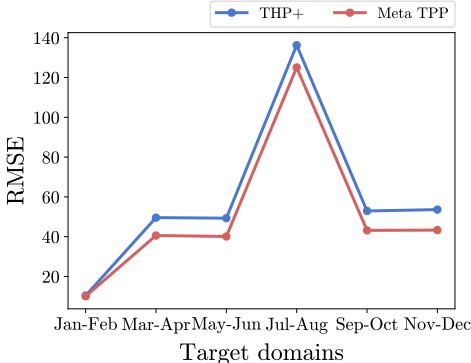

(a) Imputation with different drop rates  (b) Distribution drift in NYC Taxi dataset

Figure 2: Experiment results on imputation and distribution drift.

| Attention | Latent | Reddit | | | Uber | |
|---|---|---|---|---|---|---|
| | | RMSE | NLL | Acc | RMSE | NLL |
| ✗ | ✗ | 0.16 | 0.92 | 0.59 | 63.71 | 3.68 |
| ✗ | ✓ | 0.13 | **-0.39** | **0.61** | 63.35 | 3.25 |
| ✓ | ✗ | **0.12** | 0.29 | **0.61** | 47.91 | 3.72 |
| ✓ | ✓ | **0.12** | 0.07 | 0.60 | **21.87** | **2.98** |

Table 3: Comparison of the variants of Meta TPPs

| Methods | # Params | Reddit | | |
|---|---|---|---|---|
| | | RMSE | NLL | Acc |
| THP$^+$ | 113K | 0.26 | 1.19 | **0.60** |
| | 170K | 0.29 | 0.79 | 0.59 |
| | 226K | 0.28 | 1.44 | 0.57 |
| AttnTPP | 222K | **0.12** | **0.07** | **0.60** |

Table 4: Comparison of diff. model size

**Periodic patterns.** As previously mentioned in Section 3.3, the cross-attention module is designed to capture periodic patterns by matching the local history of the current event to the local histories of the previous event times, in addition to alleviating the underfitting problem. To validate the effectiveness of the cross-attention, we experiment on the datasets with strong periodicity – Sinusoidal, Uber, and NYC Taxi (please refer to Appendix H for details). Table 2 shows that the Attentive TPP generally outperforms the state-of-the-art methods, except for RMSE on Sinusoidal. To investigate the behavior of the cross-attention, we provide an example in Figure 3a where we highlight 15 (out of 64) the most attended local history indices (in red) to predict the target event (in green) in a sequence from Sinusoidal. The dotted grey lines represent the start and end of periods. We can observe that the attention refers to the local histories with similar patterns more than the recent ones.

## 5.3 APPLICATIONS

**Imputation.** We study the robustness of Meta and Attentive TPP to noise by randomly dropping events, simulating partial observability in a noisy environment, and measuring imputation performance. For the experiment, we drop $n$ percentage of all the event times drawn independently at random per sequence on the Sinusoidal wave dataset. In Figure 2a, we report the bootstrapped imputation performance of THP$^+$, Meta TPP, and Attentive TPP, in terms of RMSE. As the drop ratio increases, RMSE increases for all three models but the gap exponentially increases. Given that the performance gap between three models on 'next event' predictions is not as large (mean RMSE – THP$^+$: 1.72, Meta TPP: 1.49, Attentive TPP: 1.45), the results shown in Figure 2a imply that the Meta and Attentive TPP are significantly more robust to the noise coming from partial observability.

**Distribution drift.** Distribution drift occurs when the distribution observed during training becomes misaligned with the distribution during deployment due to changes in the underlying patterns over time. This is a common deployment challenge in real-world systems. Figure 2b shows how THP$^+$ and Meta TPP models trained on the January-February data of the NYC Taxi dataset generalize to subsequent months. Both models show a decrease in performance, suggesting the presence of non-stationary or seasonal patterns in the data that are not captured in the training months; however, Meta TPP is comparatively more robust across all out-of-domain settings. It is also worth mentioning that although the Attentive TPP generally performs better than Meta TPP in the conventional experimental setting, it is not the case for distribution drifts. We conjecture it is because the cross-attention is designed to alleviate the underfitting problem, which results in being less robust to distribution drift.

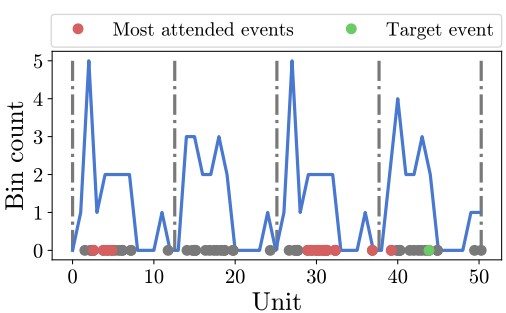

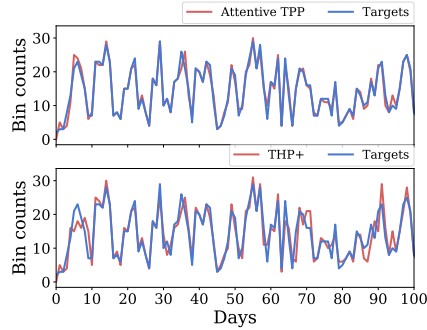

(a) Self-attention in THP and proposed cross-attention

(b) Visualization of test predictions vs. targets

Figure 3: Qualitative analysis on the cross-attention and prediction results.

## 5.4 ABLATION STUDIES

**Meta TPP and its variants.** In Table 3, we compare the proposed Meta TPP and its variants on Reddit, and Uber datasets. The result shows that both cross-attention and latent variable path generally help to improve the performance. When they are combined (resulting in the Attentive TPP), it generally performs the best in terms of both RMSE and NLL.

**Different model sizes.** Both latent path and cross-attention components introduce additional learnable parameters (for the Reddit dataset with $984$ classes, THP$^+$: 113K, Meta TPP: 126K, and Attentive TPP: 209K parameters). We provide an ablation study with varying number of model parameters for the THP$^+$ baseline to validate the performance improvement does not come from the increased number of model parameters. In Table 4, we increase the number of model parameters for THP$^+$ on Reddit along with the result of the Attentive TPP. The result shows that the larger model does not necessarily help to improve the performance: as the number of parameters increases, NLL sometimes improves but it may hurt RMSE as in the case of Table 4. The significant improvement in performance of our proposed method shows the importance of providing an effective inductive bias.

**Parameter sharing.** In Attentive NP, the encoders of the latent path and attention path are separated to provide different features. However, it can significantly increase computational overhead, and for this reason, we share the weights for the encoders. As an ablation study, we provide the performance with and without sharing the weights for the encoders of the Attentive TPP on the Stack Overflow dataset. Although the number of parameters of 'with sharing' is $50\%$ less than 'without sharing' ('without sharing': 86K vs. 'with sharing': 136K), it performs better than 'without sharing' (RMSE / NLL – 'without sharing': 1.08 / 3.20 vs. 'with sharing': 1.03 / 2.81).

**Visualization of event time predictions.** In TPP literature, the evaluation relies only on the RMSE and NLL metrics. It is, however, often hard to measure how practically useful a trained TPP model is. To qualitatively evaluate TPP models, we convert an event time sequence into time series sequence: we count the number of event times falling into each bin (in Figure 3b, each bin is a day). Figure 3b shows how close overall predictions of the Attentive TPP and THP$^+$ (in red) are to the ground truth event times (in blue). In the figure, we can see that the Attentive TPP's predictions closely align with the targets whereas the predictions of the THP$^+$ are off at some regions. Note that as the y-axis represents bin counts, even a slight off from the ground truth implies large values in terms of RMSE.

## 6 CONCLUSION

Previously, neural temporal point processes (TPPs) train neural networks in a supervised learning framework. Although performing well on event sequences similar to a training set, they are susceptible to overfitting, and may not generalize well on sequences with unseen patterns. To alleviate this, we proposed a novel framework for TPPs using neural processes (NPs). We proposed the Meta TPP where TPP is formulated as NP, and further developed the Attentive TPP using a cross-attention module, which forces a model to use similar features for repeating local history patterns. Our experiments demonstrate that the proposed models outperform strong state-of-the-art baselines on several event sequence datasets, effectively capture periodic patterns, and increase robustness to noise and distribution drift. We believe this work opens up a new research direction to meta learning for TPPs.

## REPRODUCIBILITY STATEMENT

Hyperparameters and implementation details are available in Section 5.1 and Appendix I. The code for the baselines and the proposed Meta TPP is provided in meta-tpp repo. Our code is based on publicly available official intensity-free and neural flow code.

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
