# OpenReview forum: "Meta Temporal Point Processes"
_ICLR.cc/2023/Conference — ICLR 2023 poster_

### Official Review · Reviewer_dU5L · 2022-10-20

**Confidence:** 4
**Correctness:** 3
**Technical Novelty And Significance:** 2
**Empirical Novelty And Significance:** 2
**Recommendation:** 3

**Clarity, Quality, Novelty And Reproducibility:**

Clarity: The submission provide a detailed description of the encoder part and the training procedure, but almost no description of the decoder (structure, implementation).

Novelty and Quality: The design of encoder and decoder significantly overlaps with previous works, e.g., [Zuo et al.,, 2020], [Lin et al., 2022] and [Shchur et al., 2020]. The only thing that looks new is the meta-learning framework, but this motivation is not consistent with the experimental design.

Reproducibility: I did not check the code.

**Strength And Weaknesses:**

Strength: Learning TPP in a meta learning framework is interesing and the submission provide the design of the model and training procedure. In general, this paper is easy to follow and I did not find any obvious technical errors despite of some notation problems.

Weakness: The submission provide a detailed description of the encoder part and the training procedure, but almost no description of the decoder. For example, what is the structure of the decoder? How to implement the decoder? If the output of the decoder is the distribution of the next event time or inter-event interval, what distribution did you use? and how did you parameterize the distribution with your decoder? All questions above are not answered in the submission. Furthermore, the decoder takes a random latent variable Z as input, does that mean the output distribution of the decoder is also random? If so, this kind of design contradicts with the TPP model that when the historical information is fixed, the distribution of the next event time or inter-event interval should be determinnistic.

The design of experiments is also problematic. As stated in the introduction, the goal of meta learning is to improve generalization. To make the experiments convincing, you should demonstrate the meta-learning TPP is better than existing methods on generalization, e.g., training log-likelihood v.s. test log-likelihood. The curret experiment section even did not describe how to split traning and test dataset; and looks more like the submission proposed a more expressive model with lower RMSE or NLL in Table 1,2,3. In conclusion, the experiments did not convince the statement made in the introduction.

**Summary Of The Paper:**

This work propose to train TPP in a meta learning framework and treat each sequence as a different task via the neural process. The work defines the context dataset, target input and output for TPP model. Also, the work introduces the cross-attention module to introduce local history matching to learn more informative features.


**Summary Of The Review:**

See strength and weakness.

---

> ### Author Response · Authors · 2022-11-13
> **Response to Reviewer dU5L (2/2)**
>
>
> |  |  | Sinusoidal || Uber || NYC Taxi ||
> |---|---|---|---|---|---|---|---|
> | Methods | # Params |RMSE |NLL | RMSE | NLL |RMSE |NLL |
> |THP$^+$  | 50K  | 1.72 (0.10) | 0.84 (0.02) | 90.25 (4.53) | 3.63 (0.03) | 10.31 (0.47) | 2.00 (0.01) |
> |         | 100K  | 1.84 (0.13) | 1.04 (0.02) | 82.69 (4.56) | 3.34 (0.03) | 10.16 (0.47) | **1.92 (0.01)** |
> |Attentive TPP | 96K  | **1.45 (0.11)** | **0.66 (0.02)** | **22.11 (1.94)** | **2.89 (0.04)** | **8.92 (0.42)** | 2.00 (0.009) |
>
> |  |  |  | SO | |
> |---|---|---|---|---|
> | Methods | # Params | RMSE |NLL | Acc |
> |THP$^+$ |  52K  | 1.68 (0.16) | 3.28 (0.02)  | **0.46 (0.004)** |
> |        |  103K | 1.63 (0.06)  | 2.82 (0.03)  | **0.46 (0.004)** |
> |Attentive TPP |  99K | **1.15 (0.02)** | **2.64 (0.02)** | **0.46 (0.004)** |
>
> |  |  |  | Mooc | |
> |---|---|---|---|---|
> | Methods | # Params | RMSE |NLL | Acc |
> |THP$^+$  | 56K  | 0.18 (0.005)  | 0.13 (0.02)  | 0.38 (0.004) |
> |         | 113K | 0.22 (0.007)  | 0.05 (0.03)  | **0.39 (0.004)**  |
> |Attentive TPP | 108K  | **0.16 (0.004)**  | **-0.72 (0.02)**  | 0.36 (0.003) |
>
> |  |  |  | Wiki | |
> |---|---|---|---|---|
> | Methods | # Params | RMSE |NLL | Acc |
> |THP$^+$  | 577K  | 0.17 (0.02)  | **6.25 (0.39)**  | 0.23 (0.03) |
> |         | 1153K  | 0.16 (0.01)  | 6.47 (0.40)  | 0.21 (0.02) |
> |Attentive TPP | 1149K  | **0.15 (0.01)**  | **6.25 (0.38)**  | **0.25 (0.03)** |
>
> In the table above, we observe that in many cases, smaller models perform better than larger models : on Sinusoidal, Mooc, and Wiki.
> Although it is sometimes true that larger models perform better than their smaller counterparts, they are still significantly worse than our proposed Attentive TPP.
> Note that we conducted exactly the same grid search for hyperparameter tuning for the larger models.
> The results empirically demonstrate that the improvement does not necessarily come from the size of a model but from right inductive biases.
> We also added these results in the Appendix F.
>
> Lastly, please note that all the experiment results we have reported in Table 1-4 and in rebuttal are on the test sets.
> We believe the RMSE, NLL, and Accuracy on test sets are good metrics to compare the generalization performance of different models.
> Given that our proposed method outperforms all the baselines, we think our experiments empirically demonstrate the robustness of our method in terms of generalization.
>
> ### Clarity
>
> > 1. The submission provide a detailed description of the encoder part and the training procedure, but almost no description of the decoder (structure, implementation).
>
> **Answer:**
> Please refer to the answer in the Q1 of the weakness section above.
>
>
> ### Novelty and Quality
>
> The design of encoder and decoder significantly overlaps with previous works, e.g., [Zuo et al.,, 2020], [Lin et al., 2022] and [Shchur et al., 2020]. The only thing that looks new is the meta-learning framework, but this motivation is not consistent with the experimental design.
>
> **Answer:**
> We agree that the design of the encoder and decoder are mostly the same as the previous works the reviewer mentioned.
> It is, however, intended for fair comparison.
> Our contribution is on proposing a new meta learning framework for TPPs, which has never been studied.
> By using the same encoder and decoder structures, we have tried to demonstrate that the improvement does not come from architecture but from different learning framework.
>
> To be more concrete, the proposed global feature $G$ and its latent version $z$ provide a sequence (or task) specific characteristics that are shared across all the event times in the same sequence.
> It encourages the model to treat different sequences as different tasks, which is not the case in the supervised learning setting where all sequences are treated as the realization of only one stochastic process.
> In addition, the proposed attention module further boosts the performance as *i)* it prevents the underfitting problem and *ii)* it directly attends periodic patterns in a sequence.
>
> Furthermore, as shown in the Q3 of the weakness section, our experiments on the baselines with the increased number of parameters on $6$ datasets show that the performance gain does not come from the increased number of parameters.

---

> ### Author Response · Authors · 2022-11-13
> **Response to Reviewer dU5L (1/2)**
>
> ### Weakness
>
> > 1. The submission provide a detailed description of the encoder part and the training procedure, but almost no description of the decoder. For example, what is the structure of the decoder? How to implement the decoder? If the output of the decoder is the distribution of the next event time or inter-event interval, what distribution did you use? and how did you parameterize the distribution with your decoder? All questions above are not answered in the submission.
>
> **Answer:**
> Thank you for pointing it out.
> We agree that we should have added a detailed description about the decoder.
> We provided the detailed explanation about the structure, inputs and outputs of the decoder in the Q5 of the common questions and at the end of Section 3.3.
> Please refer to them for details.
>
>
> > 2.  Furthermore, the decoder takes a random latent variable Z as input, does that mean the output distribution of the decoder is also random? If so, this kind of design contradicts with the TPP model that when the historical information is fixed, the distribution of the next event time or inter-event interval should be determinnistic.
>
> **Answer:**
> You are right about the output distribution that it is stochastic.
> But, we do not agree that it contradicts with any assumption of TPPs.
> Latent variable models and variational inference are commonly used techniques in machine learning.
> In many applications, the output probability distributions depend on the latent variable $z$ as with our work.
> For example, Lin et al. (2022) employs a diffusion model for TPPs.
> Since they also use a latent variable model, their output distribution is stochastic, and inference is done based on Monte-Carlo approximation; please refer to the last paragraph of the Page 8 in Lin et al.
>
> If the reviewer provides literature that explains the deterministic characteristics of TPP models, we are willing to check it out and modify our assumptions and models accordingly.
>
> [1] Lin et al., Exploring Generative Neural Temporal Point Process, TMLR, 2022.
>
>
>
> > 3. The design of experiments is also problematic. As stated in the introduction, the goal of meta learning is to improve generalization. To make the experiments convincing, you should demonstrate the meta-learning TPP is better than existing methods on generalization, e.g., training log-likelihood v.s. test log-likelihood. The curret experiment section even did not describe how to split traning and test dataset; and looks more like the submission proposed a more expressive model with lower RMSE or NLL in Table 1,2,3. In conclusion, the experiments did not convince the statement made in the introduction.
>
> **Answer:**
> Thank you for the constructive feedback.
> Here, we first provide data splitting scheme for train, validation, and test sets.
> For all the 7 datasets -- Stack Overflow, Mooc, Reddit, Wiki, Sinusoidal, Uber and NYC Taxi, we followed the splits made in the previous works such as Shchur et al. (2020), and Yang et al. (2022).
> More specifically, we use 60\%, 20\%, and 20\% split for train, validation, and test, respectively, for all the datasets following Shchur et al. (2020) except for Stack Overflow.
> For Stack Overflow, we follow the split made by Yang et al. (2022) and Du et al. (2016) where 4,777, 530, and 1,326 samples are assigned for train, validation, and test, respectively.
> We added the detailed description of the data splitting scheme at the beginning of Appendix I.
>
> We understand the concern that the improvement may just come from larger model size.
> Although we demonstrated that it was not the case through Table 4, we provide more evidence on the rest of the datasets as below (in the next comment).

---

> ### Author Response · Authors · 2022-11-19
> **A Reminder to Reviewer dU5L**
>
> Dear Reviewer dU5L,
>
> As a reminder, the discussion session will end in few hours.
> We appreciate your feedback, especially about lack of description for the decoder.
> We believe a detailed explanation about the decoder has been added to the responses as well as the revised version of our paper.
> Furthermore, we provided extensive experiment results that further back up the hypothesis that the improvement of our method does not come from the increased model size.
> Lastly, we argued that the stochasticity of the outputting distribution is not abnormal based on the literature in variational inference as well as TPPs.
>
> If there are still any doubts or concerns, please let us know so that we can address them.
> Also, your feedback about our responses would be highly appreciated so please let us know if our responses are satisfying or need further clarification.
> Thank you.

---

> ### Author Response · Authors · 2022-11-25
> **A Reminder to Reviewer dU5L**
>
> Dear Reviewer dU5L,
>
> As a friendly reminder, we are waiting for your valuable feedback to our responses. We understand your concerns about clarity and lack of evidence in experiments (related to your question 3 in Weakness). We believe we have addressed all your concerns in the rebuttal as well as revised paper. So, it would be highly appreciated if you can provide any feedback about our responses. If you have any further concerns, questions, or suggestions, we are willing to discuss and reflect them to the revision.
> We look forward to hearing from you. Thank you.

---

> ### Author Response · Authors · 2022-12-01
> **A Reminder to Reviewer dU5L**
>
> Dear Reviewer dU5L,
>
> We are sending you a reminder as the deadline for discussion is due this month.
> Again, we are happy to answer any further questions or concerns. Your feedback would be highly valuable for us to make this work better. We look forward to hearing from you. Thank you.

---

> ### Author Response · Authors · 2022-12-10
> **A Reminder: Discussion Ends in 3 Days**
>
> Dear Reviewer dU5L,
>
> As the deadline for the discussion is fast approaching, we want to ensure that all of your concerns have been addressed. Since the other reviewers have increased their scores, indicating that they are satisfied with our responses, we believe that our responses have also addressed your concerns. We would greatly appreciate any feedback you may have regarding our responses. Thank you for your time and consideration.

---

> ### Author Response · Authors · 2022-12-12
> **A Reminder: Discussion Ends in Few Hours**
>
> Dear Reviewer dU5L,
>
> We have only few hours for discussion and have not yet heard back from you. We are unsure if your score based on the draft is still valid for our revised paper.
>
> Regardless, we would like to know if we have addressed all of your comments and concerns in our revised submission. If you have any further feedback, please let us know as soon as possible so that we can incorporate it into our final paper.
>
> We value your input and appreciate the time and effort you have put into reviewing our work. Thank you.

---

### Official Review · Reviewer_AxMo · 2022-10-24

**Confidence:** 4
**Correctness:** 4
**Technical Novelty And Significance:** 3
**Empirical Novelty And Significance:** 3
**Recommendation:** 6

**Clarity, Quality, Novelty And Reproducibility:**

Several definitions and concepts need clarification
- As mentioned, the meta learning learns the distribution of label $y$, $p_\theta(y | x, C)$. It is unclear how $p_\theta$ is specified in this study, especially in (4). What do the authors mean by "learn" the distribution in contrast to that it appears also in the supervised learning objectives. What's the difference?
- What's the output of the decoder? According to the preliminaries, it's a point estimate/prediction of the label s.t. $f_{\theta}(\cdot, C): x \rightarrow y$. However, it's drew like a distribution in Fig 1.
- The text says, "In inference, as we do not have access to $C_L$ at $i$-th event when $l < L$, we use $z$ from $p_θ(z | C_l)$." Following this, would the KL term in (7) become $0$?
- The model involves latent variable. Sampling relies on its posterior. The how is the testing done for a trained model on new sequences as you don't have the posterior for them.
- Information about how RMSE and NLL calculated is needed, like what likelihood is used, how point prediction is obtained if the output is a distribution.

Quality:
- It's better to show the performance of a null model along with the other methods, e.g., the ones using median interevent interval to predict future event time. It's possible, even not expectedly, that no methods do well for certain dataset.
- It's better to describe how the error bar is calculated in the text or caption.




**Strength And Weaknesses:**

Strength:
Use meta learning training to overcome the generalization issue of supervised learning.
The latent variable layer incorporates uncertainty.
The attention takes into account the similarity between inputs.

Weaknesses:
The meta objective (4) turns out a special case of supervised learning (3). The distinction is blurry.
The writing needs to be improved.

**Summary Of The Paper:**

In this work, the authors propose to use a meta learning way to train neural processes aiming at one-step prediction of event time of point processes. The proposed TPPs employ a latent variable layer and attention mechanism to improve the prediction performance. The method was demonstrated with benchmark datasets.

**Summary Of The Review:**

This work uses a meta learning to train variant TPP aiming at one-step prediction of event time of point processes. The demonstration with a variety of benchmark datasets shows competitive performance against other SOTA methods. There are a few key concepts and details need to be clarified to make the work a better understandable and solid one.

---

> ### Author Response · Authors · 2022-11-13
> **Response to Reviewer AxMo (2/2)**
>
>
> > 4. The model involves latent variable. Sampling relies on its posterior. The how is the testing done for a trained model on new sequences as you don't have the posterior for them.
>
> **Answer:**
> Suppose we want to predict the distribution of the event time $\tau\_{l+1}$ given the previous event times: $p\_\theta( \tau\_{l+1} \\, | \\, \tau\_{\leq l})$.
> In the proposed Meta TPP framework, it is converted into $p\_\theta( \tau\_{l+1} \\, | \\, \tau\_{l-k+1:l}, \mathcal{C}\_l)$.
> In inference, we do not have access to $\mathcal{C}\_L$ but can still sample from $p\_\theta (z \\, | \\, \mathcal{C}\_l)$.
> As shown in the question 3, we can approximate the NLL using the samples $z\_m \sim p\_\theta (z \\, | \\, \mathcal{C}\_l)$.
> % We provided how to compute the RMSE and accuracy in the question 5.
>
> > 5. Information about how RMSE and NLL calculated is needed, like what likelihood is used, how point prediction is obtained if the output is a distribution.
>
> **Answer:**
> We agree that the computation of NLL and RMSE was not clear. We provided the detailed explanation about how evaluation metrics -- NLL, RMSE, and Accuracy, are computed in the Q4 of the common questions as well as Appendix C. Please refer to them for detailed information.
>
> ### Quality
>
> > 1. It's better to show the performance of a null model along with the other methods, e.g., the ones using median inter-event interval to predict future event time. It's possible, even not expectedly, that no methods do well for certain dataset.
>
> **Answer:**
> Thank you for a valuable suggestion.
> We also considered to have a naive baseline but decided not to have them in the paper, based on a reasoning provided in Shchur et al. (2021).
> In Section 7.2 of Shchur et al. (2021), the authors argue that the metrics such as mean absolute error (MAE) or mean squared error (MSE) can be less suitable than NLL since one can train a baseline model that only produces point estimation for next event times $\tau$.
> This baseline can be trained directly minimizing MAE or MSE.
> But since TPPs are probabilistic models trained with NLL, comparing the baseline and regular TPP models on the above metrics is unfair.
> As their point makes sense and the baseline model (including the naive baseline) cannot provide the performance in NLL, we did not add any naive baselines.
>
> But as suggested, we implemented a naive baseline that makes predictions based on median inter-event interval: $\hat{\tau}\_{l+1} = \tau\_{l} + \Delta \tau\_{median, l}$ where $\Delta \tau\_{median,l}$ is a median of the inter-event interval up to $l$-th event.
> We boostrapped for 200 times on the test set to obtain the mean of RMSE metrics as with how we obtained the numbers for the other methods (NLLs are not available for the naive baseline).
>
> | Methods        | Stack Overflow | Mooc | Reddit | Wiki | Sinusoidal | Uber | NYC Taxi |
> |:---:|:---:|:---:|:---:|:---:|:---:|:---:|:---:|
> | Naive baseline |   161.21  |  0.79   |  0.38   |  0.21    |  4.61   |   107.91  |  24.58   |
> | Intensity-free |  3.64    |  0.31   | 0.18    |  0.60 |   1.29  |   51.23   |  46.59   |
> | Neural flow    |  --    |  0.47   |  0.32   |  0.56    |  **1.13**   |  --   |   --  |
> | THP$^+$        |  1.68    | 0.18    |  0.26   | 0.17     |  1.72   |  90.25   |  10.31   |
> | Attentive TPP  |  **1.15**    |  **0.16**   |  **0.11**   |  **0.15**     |  1.45   |  **22.11**   |  **8.92**   |
>
> The performance of the naive baseline is surprisingly good for some cases.
> For instance, it is better than the intensity-free on Wiki and NYC Taxi datasets.
> It is, however, much worse than THP$^+$ and the proposed Attentive TPP on all the datasets.
> We added the description and results above in Appendix E.
>
>
>
> > 2. It's better to describe how the error bar is calculated in the text or caption.
>
> **Answer:**
> As we mentioned in the **Hyperparmeters** in Section 5.1, we boostrapped 200 times on the test set to approximate the mean and standard deviation following Yang et al. (2022).
> More specifically, we sampled the test set with replacement for the same size as the original test set to obtain evaluation metrics, and repeated for 200 times to get mean and standard deviation.
> To make it easier to find the description in the revision, we added pointers to the description for each table and figure where error bars are used.

---

> > ### Comment · Reviewer_AxMo · 2022-11-19
> > **Response**
> >
> > Thank the authors for the responses. I have no further questions.

---

> > > ### Author Response · Authors · 2022-11-19
> > > **Response to Reviewer AxMO**
> > >
> > > Thank you for the response.
> > > Can we ask what are specific reasons that you think this work is still below the acceptance threshold?
> > > Are there any points that we failed to clarify?
> > > We are willing to address any doubts or questions you have, and revise our paper accordingly.
> > > Any feedback would be helpful to improve our work. So please feel free to let us know. We look forward to hearing from you.

---

> > > > ### Comment · Reviewer_AxMo · 2022-11-19
> > > > **Response to Authors**
> > > >
> > > > I have increased the score.

---

> > > > > ### Author Response · Authors · 2022-11-19
> > > > > **Response to Reviewer AxMO**
> > > > >
> > > > > Dear Reviewer AxMO,
> > > > >
> > > > > Thank you for your feedback. We will make sure all your suggestions are reflected to the revision. Also, please feel free to let us know if you need any further clarification during AC-Reviewer discussion session.

---

> ### Author Response · Authors · 2022-11-13
> **Response to Reviewer AxMo (1/2)**
>
> ### Weaknesses
>
> > 1. The meta objective (4) turns out a special case of supervised learning (3). The distinction is blurry.
>
> **Answer:**
> We agree that the objective of the proposed Meta TPP in Equation (4) is a special case of the objective of the general TPP in Equation (3).
> But, we believe that it is a strength not a weakness.
> In general, meta learning algorithms are not directly comparable to a supervised learning counterpart since meta learning requires an additional context set in test time.
> In this work, we deliberately avoided it, and used the same information $\tau\_{\leq l}$ as with the previous TPP methods under the supervised learning framework.
>
> For the design of a deep learning algorithm, it is important to inject right inductive biases, and in our case, we provided the global latent variable $z$ and attention feature $r\_l^\prime$ as inductive biases.
> Please refer to the Q2 in the common questions for detailed explanations about the inductive bias from the global latent variable $z$.
> Although we use the exact same information $\tau\_{\leq l}$ in the end, our proposed methods significantly outperforms the baselines because of the right guidance.
> Note that we use the same encoder and decoder as the THP$^+$ baseline.
> Furthermore, Table 4 shows that improvement in performance does not come from the increased model parameters.
> In Appendix F (and the answer to the Q3 in weakness under Reviewer dU5L), we provide further experiment results that support the findings from Table 4.
>
>
>
>
> > 2. The writing needs to be improved.
>
> **Answer:**
> Thank for pointing it out.
> We fixed all the questions and vague statements raised below, and reflected them to the revision.
> Please check them out and let us know if there is any further improvement we can make.
>
>
> ### Clarity, Quality, Novelty And Reproducibility
>
> > 1. As mentioned, the meta learning learns the distribution of label y, $p\_\theta (y \\, | \\, x, \mathcal{C})$. It is unclear how $p\_\theta$ is specified in this study, especially in (4).
> What do the authors mean by "learn" the distribution in contrast to that it appears also in the supervised learning objectives. What's the difference?
>
> **Answer:**
> This is a subtle but important point.
> We should have made it clearer in the main paper.
> We explained this point in detail in the Q2 of the common questions above, and reflected in the Appendix B. Please refer to them.
>
>
>
> > 2. What's the output of the decoder? According to the preliminaries, it's a point estimate/prediction of the label s.t. $f\_\theta (\cdot, \mathcal{C}): x \rightarrow y$. However, it's drew like a distribution in Fig 1.
>
> **Answer:**
> The point estimation mapping: $f\_\theta (\cdot, \mathcal{C}): x \rightarrow y$, represents the function learned in a ``regular'' meta learning setting.
> In the paragraph where the Neural Processes are introduced in preliminaries, we specified that the mapping that Neural Process models learn is $p\_\theta (y \\, | \\, x, \mathcal{C})$.
> The proposed Meta and Attentive TPP also learns the same mapping.
>
> We added detailed explanation about the decoder -- input features, outputs to determine the probability distribution of the next event time $\tau\_{l+1}$, and structure of it, in the Q5 of the common questions, and at the end of Section 3.3, so please refer to them for more details.
>
>
>
> > 3. The text says, "In inference, as we do not have access to $\mathcal{C}\_L$ at $i$-th event when $l < L$, we use z from $p\_\theta (z \\, | \\, \mathcal{C})$." Following this, would the KL term in (7) become 0?
>
> **Answer:**
> Thank you for pointing it out. We understand where the confusion comes from.
> In inference, our goal is no longer computing the variational lower bound: we want to compute evaluation metrics such as NLL, RMSE, and accuracy.
>
> For NLL, we want to compute $\log p\_\theta (\tau\_{l+1}  \\, | \\,  \tau\_{l-k+1:l}, \mathcal{C}\_l)$.
> With the latent variable, it can be approximated using Monte-Carlo (MC) approximation as follows,
> \begin{align}
> \log p\_\theta (\tau\_{l+1}  \\, | \\,  \tau\_{l-k+1:l}, \mathcal{C}\_l) &= \log \int p\_\theta (\tau\_{l+1}  \\, | \\,  \tau\_{l-k+1:l},z) p\_\theta (z  \\, | \\,  \mathcal{C}\_l) dz \\\\
> &\approx \log \frac{1}{M} \sum\_{m=1}^M p\_\theta (\tau\_{l+1}  \\, | \\,  \tau\_{l-k+1:l},z_m)
> \end{align}
> where $z\_m \sim p\_\theta ( z  \\, | \\,  \mathcal{C}\_l)$.
> Note that here, we sample $z$ from $p\_\theta ( z  \\, | \\,  \mathcal{C}\_l)$ which is trained to be similar to $p\_\theta (z  \\, | \\,  \mathcal{C}\_L)$ since we no longer have access to $\mathcal{C}\_L$ in inference, which is what we meant by the sentence you quoted (for the other evaluation metrics, please refer to the Q4 in the common questions).
>
> To make this distinction clearer, we modified the explanation about Equation (5) to (7) so that it reflects the above description in the revision.

---

### Official Review · Reviewer_YAR9 · 2022-10-25

**Confidence:** 3
**Correctness:** 4
**Technical Novelty And Significance:** 3
**Empirical Novelty And Significance:** 3
**Recommendation:** 6

**Clarity, Quality, Novelty And Reproducibility:**

The paper is largely well written. Some terms/notations could be defined more precisely (see Comments above).
The presentation could be more self-contained at some points, e.g., by not just referring to the transformer point process models.


**Strength And Weaknesses:**

Strengths:
- The meta-learning framework is new for point process models, as far as I am aware.
- The method appears to perform better than point process models trained in a supervised manner.

Weaknesses:
- I feel that the presentation could be improved at some points (see below).

Actionable feedback:
- The equations (5) to (7) are a bit unclear to me. Why are we using a variational lower bound based on an expectation with respect to $p_{\theta}(z|C_{L})$? Is this still a valid lower bound if we take (as I understand you do, while dropping the KL term?) using samples from $p_{\theta}(z|C_{l})$?
- How are the context features $r_1, r_2,$ … computed/encoded?
- How exactly does z enter the decoder. What are the priors for z given different context sets?

Comments:
- Can this be extended to multi-dimensional/marked point process setting where there is not just a single task to predict?
- I was wondering if the presence of periodic patterns in the data somewhat makes the required permutation invariance over a context set a constraint that might decrease the forecasting performance?
- Can this method be applied to meta-learning short sequences, e.g., setups like in Xie et al. Meta Learning with Relational Information for Short Sequences, Neuirps 2019?




**Summary Of The Paper:**

The authors suggest viewing the problem of predicting the next event for temporal point process models as a meta-learning problem. In particular, they advocate a neural process framework, whereby windows of previous event times act as context and target input sets. A cross-attention architecture is suggested to increase modeling capacity that does not scale quadratically in the number of events. The method seems to outperform supervised point process models empirically.


----
Update following authors' responses:
The presentation and clarity has been improved in the updated manuscript. The authors have also included additional experimental work that improved the experimental validation of the proposed method. I have increased my score to a weak accept.

---

**Summary Of The Review:**

Casting predictions with point process as a neural process is new, as far as I am aware. I feel there is scope to improve the presentation and clarity and I would consider increasing my score if this were addressed.

---

> ### Author Response · Authors · 2022-11-13
> **Response to Reviewer YAR9 (2/2)**
>
> ### Comments
>
> > 1. Can this be extended to multi-dimensional/marked point process setting where there is not just a single task to predict?
>
> **Answer:**
> Thank you for sharing the interesting directions for future work.
> If the reviewer meant modeling multiple future event times for TPP by multi-dimensional, we do not think it has been explored in any of previous works including ours yet (please let us know if our interpretation is wrong).
> It is, however, highly interesting and potentially important research direction in the field.
>
> For marked point processes, we extended the proposed method to the marked cases by adding a class prediction branch.
> We added the detailed description to the Q3 of the common questions above, and Appendix G.
> Please refer to them.
>
>
>
> > 2. I was wondering if the presence of periodic patterns in the data somewhat makes the required permutation invariance over a context set a constraint that might decrease the forecasting performance?
>
> **Answer:**
> Thank you for sharing an interesting research question.
> According to our experiments including the results of the THP$^+$ baseline in Table 2 and Meta TPP in Table 5 in the Appendix, we observe that having the permutation invariant global feature still generally helps to improve the performance even on the datasets with strong periodic patterns.
>
> Here, we compare the THP$^+$ baseline and Meta TPP on Sinusoidal, Uber and NYC Taxi datasets (they all have strong periodicity).
> The decoder of the Meta TPP takes the global latent feature $z$ (from the permutation invariance constraint) as an input, in addition to the target input feature $r\_l$ that the decoder of the THP$^+$ baseline takes as input.
>
> |  | Sinusoidal || Uber || NYC Taxi ||
> |:---:|:---:|:---:|:---:|:---:|:---:|:---:|
> | Methods |RMSE |NLL | RMSE | NLL |RMSE |NLL |
> |THP$^+$ | 1.72 | 0.84  | 90.25 | 3.63 | 10.31 | **2.00** |
> |Meta TPP | **1.48** | **0.61** | **63.35** | **3.25** | **10.04** | 2.33 |
>
> The permutation invariance constraint and the resulting feature $z$ can help predicting the next time step because the global feature is designed to capture the global and sequence-specific characteristics, which helps adapting to a new unseen sequence (or task).
> But it is not the case for previous TPP methods where there is no sequence-specific features.
> We explained what the motivation of the global feature and how it is learned in the Q2 of the common questions above, and the Appendix B.
>
>
> > 3. Can this method be applied to meta-learning short sequences, e.g., setups like in Xie et al. Meta Learning with Relational Information for Short Sequences, Neuirps 2019?
>
> **Answer:**
> Thank you for sharing an interesting work.
> If our understanding is correct about the paper, there are two major differences between Xie et al. (2019) and our work -- *i)* Xie et al. employ an optimization-based meta learning framework such as MAML to learn the global parameters whereas ours employ a model-based meta learning with stochasticity such as Neural Processes, and *ii)* Xie et al. is tailored to settings where additional graph information is available and can be leveraged to improve the prediction of short sequences.
>
>
> But, without taking into account the graph information (which is included in the datasets used in Xie et al.), our proposed methods are applicable to on event time sequence data including short sequences.
> For instance, we conducted additional experiments on MIMIC-II dataset (Johnson et al., 2016), one of popular TPP datasets, of which statistics are as follows,
>
> | # of Sequence | # of Events | Max Seq. Length | # of marks | Avg. Seq. Length |
> | :---: | :---: | :---: | :---: | :---: |
> | $650$ | $2,419$ | $33$ | $75$ | $3.72$ |
>
> Note that the average sequence length of MIMIC-II is even smaller than any of the datasets used in Xie et al. (2019); LinkedIn: $4.9$, MathOverflow $11.8$, and StackOverflow: $7.7$.
> Hence, MIMIC-II is in the regime of short sequences.
> The experiment results of the baselines and our method on the MIMIC-II dataset are as below.
> The results imply that our proposed method outperforms the baselines even on short sequence datasets like MIMIC-II.
>
> |  |   | MIMIC-II | |
> |:---:|:---:|:---:|:---:|
> | Methods        | RMSE | NLL | Acc |
> | Intensity-free |  1.40 (0.22) | 1.20 (0.24) | **0.85 (0.05)**     |
> | Neural flow    |  1.52 (0.24) | 2.00 (0.27) | 0.63 (0.04)    |
> | THP$^+$        |  1.55 (0.21) | 1.24 (0.20) | 0.83 (0.05)    |
> | Attentive TPP  |  **1.25 (0.15)** | **1.09 (0.29)** | **0.85 (0.05)**    |
>
> [1] Johnson et al., MIMIC-III, a freely accessible critical care database, Scientific Data, 2016.

---

> ### Author Response · Authors · 2022-11-13
> **Response to Reviewer YAR9 (1/2)**
>
> ### Actionable Feedback
>
> > 1. The equations (5) to (7) are a bit unclear to me. Why are we using a variational lower bound based on an expectation with respect to $p\_\theta (z\\, | \\,\mathcal{C}\_L)$? Is this still a valid lower bound if we take (as I understand you do, while dropping the KL term?) using samples from $p\_\theta (z\\, | \\,\mathcal{C}\_l)$?
>
> **Answer:** We understand the confusion. Let us clarify the variational lower bound in Equation (5) to (7) for training and inference, separately.
>
> In training time, we maximize the variational lower bound in Equation (6), defined as,
> $E\_{z \sim p\_\theta(z \\, | \\, \mathcal{C}\_L)} \log p\_\theta(\tau\_l  \\, | \\,  \tau\_{l-k:l-1}, z) - KL(p\_\theta (z  \\, | \\,  \mathcal{C}\_L) \\;||\\; p\_\theta (z  \\, | \\,  \mathcal{C}\_l))$.
> Since we have access to the whole sequence in training time, it is valid to sample $z$ from the posterior distribution $p\_\theta (z \\, | \\, \mathcal{C}\_L)$, from which we can compute the Equation (7):
> $\frac{1}{N} \sum\_{n=1}^N \log p\_\theta (\tau\_l  \\, | \\,  \tau\_{l-k:l-1}, z_n)- KL(p\_\theta (z  \\, | \\,  \mathcal{C}\_L) \\;||\\; p\_\theta (z  \\, | \\,  \mathcal{C}\_l))$.
> The reason why we use $p\_\theta (z  \\, | \\,  \mathcal{C}\_L)$ as the inference network for the amortized variational inference is that $z \sim p\_\theta (z  \\, | \\,  \mathcal{C}\_L)$ contains all the information of the whole sequence.
> Due to the KL divergence term, regardless of the target index $l$, it always minimizes the distance between $p\_\theta (z  \\, | \\,  \mathcal{C}\_L)$ and $p\_\theta (z  \\, | \\,  \mathcal{C}\_l)$.
> It implies that the latent variable $z \sim p\_\theta (z \\, | \\, \mathcal{C}\_l)$ is guided to capture the global feature of the whole sequence, which is equivalent to $z \sim p\_\theta (z  \\, | \\,  \mathcal{C}\_L)$.
>
> In inference time, we do not compute the variational lower bound.
> Here, our goal is to compute the evaluation metrics such as NLL, RMSE and accuracy. For NLL, we want to compute $\log p\_\theta (\tau\_l  \\, | \\,  \tau\_{l-k:l-1}, \mathcal{C}\_l)$.
> With the latent variable, it can be approximated as follows,
> \begin{align}
> \log p\_\theta (\tau\_l  \\, | \\,  \tau\_{l-k:l-1}, \mathcal{C}\_l) &= \log \int p\_\theta (\tau\_l  \\, | \\,  \tau\_{l-k:l-1},z) p\_\theta (z  \\, | \\,  \mathcal{C}\_l) dz \\\\
> &\approx \log \frac{1}{M} \sum\_{m=1}^M p\_\theta (\tau\_l  \\, | \\,  \tau\_{l-k:l-1},z_m)
> \end{align}
> where $z\_m \sim p\_\theta ( z  \\, | \\,  \mathcal{C}\_l)$.
> Note that here, we sample $z$ from $p\_\theta ( z  \\, | \\,  \mathcal{C}\_l)$ which is trained to be similar to $p\_\theta (z  \\, | \\,  \mathcal{C}\_L)$.
> We believe the confusion is coming from the misunderstanding on the goal of inference; we want to compute evaluation metrics such as NLL instead of variational lower bound in inference.
> We explained in detail how other evaluation metrics are computed in inference in the Q4 of the common questions above, and Appendix C so please refer to them.
> Also, to make this distinction clearer, we modified the explanation about Equation (5) to (7) in the revision.
>
>
>
> > 2. How are the context features $r\_1, r\_2$ computed/encoded?
>
> **Answer:**
> Please refer to the Q1 of the common questions above.
>
>
>
> > 3. How exactly does $z$ enter the decoder. What are the priors for $z$ given different context sets?
>
> **Answer:**
> We provided the detailed information about the decoder in the Q5 of the common questions, and at the end of Section 3.3.
> Please refer to them for how $z$ enters the decoder.
>
> For the last question, we assume that the reviewer refer to "the priors for z given different context sets" as $p\_\theta (z \\, | \\, \mathcal{C}\_l)$ for $l \in [1, L-1]$.
> Following the convention in Neural Processes literature and many other works using variational inference, we assume that $z$ is sampled from a Gaussian distribution.
> As mentioned in Section 3.2, with this assumption, we can use reparametrization trick as $z = \mu + \sigma \odot \epsilon$ where $\mu$ and $\sigma$ are learned from a neural network.
> Hence, given different context sets, $p\_\theta (z \\, | \\, \mathcal{C}\_l)$ changes to different Gaussian distributions depending on $\mu$ and $\sigma$.
>
> If our understanding of your question is incorrect, please feel free to let us know.

---

> ### Author Response · Authors · 2022-11-19
> **A Reminder to Reviewer YAR9**
>
> Dear Reviewer YAR9,
>
> As a reminder, the discussion session will end in few hours.
> We believe we have addressed all your suggestions about writing including unclarity of Equation (5) to (7), the encoder as well as the decoder.
> In addition, we added further experiments on the short sequences inspired by Xie et al. that you introduced.
> If you have any further doubts or suggestions, we are willing to answer them until the last minutes.
> So, please let us know if our responses are satisfying or need more revision.
> We look forward to hearing from you. Thank you.

---

> ### Author Response · Authors · 2022-11-25
> **A Reminder to Reviewer YAR9**
>
> Dear Reviewer YAR9,
>
> As a friendly reminder, we are waiting for your valuable feedback to our responses. In your summary of review, we appreciate recognizing the novelty of our work, and understand the concerns about clarity and presentation. We believe they have been addressed in the rebuttal and revised paper, so please check them out and let us know if you have any remaining questions or concerns. We are more than happy to discuss any further issues. We look forward to hearing from you. Thank you.

---

> ### Author Response · Authors · 2022-12-01
> **A Reminder to Reviewer YAR9**
>
> Dear Reviewer YAR9,
>
> We are sending you a reminder as the deadline for discussion is due this month.
> Again, we are happy to answer any further questions or concerns. Your feedback would be highly valuable for us to make this work better. We look forward to hearing from you. Thank you.

---

### Official Review · Reviewer_Jm5S · 2022-10-28

**Confidence:** 4
**Correctness:** 3
**Technical Novelty And Significance:** 3
**Empirical Novelty And Significance:** 3
**Recommendation:** 8

**Clarity, Quality, Novelty And Reproducibility:**

Clarity & reproducibility: While the experimental setup is described very precisely (except the parts mentioned above), some aspects of the model architecture are not clear.

Novelty: The connection between TPPs and meta-learning has not been explored in earlier works, as far as I am aware.

**Strength And Weaknesses:**

- Strong performance: The proposed approach shows a consistent improvement over the baselines in terms of event prediction capabilities.
- Extensive empirical evaluation: The experiments cover a large selection of datasets and tasks. These include both standard results on predictive performance (event time prediction, log-likelihood), as well as experiments designed to shed light on the properties of different methods (effect of missing events, distribution shifts on performance; effect of model size)


Weaknesses:
- Clarity: Some aspects of the proposed approach are hard to understand or are not described in full detail. The paper would benefit greatly if these are clarified in the revised version.
    - It would be helpful to explain the mapping between the variables $\mathcal{X}, \mathcal{Y}$ in the meta-learning problem definition and $\tau_i$ in the TPP problem definiton early in the paper, and then stick to the latter notation. Currently, section 3.1 is somewhat hard to follow with the switches between two notations.
    - How are the context features $r$ defined for TPP models?
    - Most notation in figure 1 ($r$, $G$) is only introduced a few pages later in the paper, making it hard to interpret the figure the first time it's encountered.
    - What is the interpretation of the global feature $G$ and the latent variable $z$ in the context of TPP modeling (section 3.2)?
    - Experiments in table 1 include accuracy scores (probably, for mark prediction?), but the rest of the paper only discusses unmarked TPPs.
    - $G$ denotes the context set in the beginning of page 4 instead of the $\mathcal{C}$ notation.
    - How is the NLL computed for the proposed Attentive TPP model? Equation 5 seems to imply that the NLL is intractlible, and only ELBO is available.
    - How exactly are the histograms in figure 3 obtained? As far as I understand, the models produce a one-step-ahead prediction of the inter-event times. How are these converted to the histograms? Do we predict the next inter-event time $\tau_{i+1}$ and then feed in the actually observed value of $\tau_{i+1}$ to then generate the prediction for $\tau_{i+2}$, etc.?

----
Post-rebuttal update: The authors have addressed all of my concerns and updated the paper accordingly. I have raised my score to reflect this.

**Summary Of The Paper:**

The paper proposes a meta-learning framework for temporal point processes (TPPs). The paper points out a limitation of the existing TPP models - treating all event sequences in the dataset as realization of the same process. The proposed meta-learning approach addresses this limitation and leads to improved performance compared to other neural TPP models, as shown by the experiments. More specifically, the proposed model uses a cross-attention encoder (based on [attentive neural process](https://arxiv.org/abs/1901.05761)) to extract the necessary information for predicting the tiem of the next event in the sequence.

**Summary Of The Review:**

The proposed approach is well-motivated and achieves strong empirical results. However, some important aspects of the model are not described very clearly.

---

> ### Author Response · Authors · 2022-11-13
> **Response to Reviewer Jm5S (2/2)**
>
>
> > 7. G denotes the context set in the beginning of page 4 instead of the $C$ notation.
>
> **Answer:** We appreciate for capturing the typo. We modified the wrong notations for $\mathcal{C}$ (misspelled to  $G$) in the beginning of page 4.
>
> > 8. How is the NLL computed for the proposed Attentive TPP model? Equation 5 seems to imply that the NLL is intractlible, and only ELBO is available.
>
> **Answer:** It is true that it is intractable to compute NLL in Equation (5) analytically. In inference, since we can obtain much larger samples for the latent variable $z$ without slowing down the inference much, we can closely approximate the NLL using the Monte-Carlo (MC) approximation.
> We explained in detail how evaluation metrics are computed in inference in the Q4 of the common questions, and Appendix C. Please refer to them.
>
> > 9. How exactly are the histograms in figure 3 obtained? As far as I understand, the models produce a one-step-ahead prediction of the inter-event times. How are these converted to the histograms? Do we predict the next inter-event time  $\tau\_{i+1}$ and then feed in the actually observed value of $\tau\_{i+1}$ to then generate the prediction for $\tau\_{i+2}$
>
> **Answer:**
> As you understood, a model produces one-step-ahead predictions of the next event times given all the previous (groundtruths) event times.
> In other words, $\hat{\tau}\_{i+1}$, a prediction at time $i+1$, is obtained using all the previous (groundtruths) event times $\tau\_{1:i}$.
> Once we have a sequence of predictions $\{\hat{\tau}\_2, \hat{\tau}\_3, \cdots, \hat{\tau}\_L \}$ (we make predictions from $2$nd event time), we can generate histograms like the ones in Figure 3.
>
> As we described in the **Visualization of event time predictions** paragraph in Section 5.4, we converted a sequence of event time predictions into times series sequence by counting the number of event times falling into each bin, say a day for unit. For instance, if a sequence of predictions are $[1.3, 8.5, 17.2, 25.6, 40.5 ]$ in hours, the bin counts for two bins $[0 \sim 24], [24 \sim 48]$ are $[3, 2]$.
> In this way, the visualization shows the overall offsets between the groundtruths and predictions of event times in macroscopic perspective of view that neither NLL nor RMSE metrics can provide.
> To the best of our knowledge, it has not been exploited in TPP literature.

---

> > ### Comment · Reviewer_Jm5S · 2022-11-21
> > **Response from the reviewer**
> >
> > Thank you for the detailed response, it addresses most of my concerns. One remaining question that I have is how the distribution $p(z | \mathcal{C}_l)$ is obtained for the held-out sequence from the test and validation sets. This seems quite important for the final evaluation of the approach.

---

> > > ### Author Response · Authors · 2022-11-21
> > > **Response to Reviewer Jm5S**
> > >
> > > We are glad to hear that most of your concerns have been addressed.
> > > We also appreciate your constructive feedback to make this work better.
> > > For your remaining question, here is our answer.
> > >
> > > As in conventional amortized variational inference, we assume that $p(z \\;|\\; \mathcal{C}\_l)$ is a multivariate Gaussian distribution with a diagonal sigma, $\mathcal{N}(z; \mu, Diag(\sigma^2))$.
> > > Hence, to define $p(z \\;|\\; \mathcal{C}\_l)$, we just need to know its mean $\mu$ and sigma $\sigma$.
> > > In training time, a model learns a mapping between the global feature $G$ and mean as well as sigma using functions $\mu(G)$ and $\sigma(G)$, respectively, where $G$ is the global feature from the encoder $Enc\_\theta$.
> > > Here, $\mu(\cdot)$ and $\sigma(\cdot)$ consist of two fully connected layers, respectively.
> > >
> > > In inference time (on both validation and test sets), given $\mathcal{C}\_l = \\{ \tau\_{t-k+1:t} \\}\_{t=1}^{l-1} $ from a sequence, we first obtain the global feature $G = Avg(r\_1, r_2, \dots r\_{l-1})$ where $r\_t = Enc\_\theta(\tau\_{t-k+1:t})$.
> > > We then compute $\mu(G)$ and $\sigma(G)$ from which we obtain $p(z \\;|\\; \mathcal{C}\_l) = \mathcal{N}(z; \mu(G), Diag(\sigma^2(G)))$.
> > >
> > > We hope the above explanation answers your question.
> > > Since we cannot update our manuscript at this point, we will reflect the above description to the revision in the future.

---

> > > > ### Comment · Reviewer_Jm5S · 2022-11-22
> > > > **Response from the reviewer**
> > > >
> > > > Thank you for the clarification, I have updated my score.

---

> > > > > ### Author Response · Authors · 2022-11-22
> > > > > **Response to Reviewer Jm5S**
> > > > >
> > > > > We appreciate recognizing the value of our work and updating the score accordingly.
> > > > > We will make sure all the concerns you raised are reflected in the revision. Thank you for your constructive feedback.

---

> ### Author Response · Authors · 2022-11-13
> **Response to Reviewer Jm5S (1/2)**
>
> ### Weaknesses
>
> > Q1. Clarity: Some aspects of the proposed approach are hard to understand or are not described in full detail. The paper would benefit greatly if these are clarified in the revised version.
>
> **Answer:** Thank you for your constructive feedback.
> We agree that it could have been better with more details about some components.
> We tried to clarify all the concerns you had below and reflected them in the revision.
> The revised parts are added in blue to the updated pdf.
>
>
>
> > Q2. It would be helpful to explain the mapping between the variables $\mathcal{X}, \mathcal{Y}$ in the meta-learning problem definition and $\tau_i$ in the TPP problem definiton early in the paper, and then stick to the latter notation. Currently, section 3.1 is somewhat hard to follow with the switches between two notations.
>
>
> **Answer:** We understand the confusion.
> The reason why we use $\mathcal{X}$ and $\mathcal{Y}$ notations is that it is hard to explain the conditions for general NPs using $\tau\_i$ notations because in TPP, a target input $\tau\_{l-k+1:l}$ (corresponding to $\mathcal{X}$) and target label $\tau\_l$ (corresponding to $\mathcal{Y}$) are not sets whereas $\mathcal{X}$ and $\mathcal{Y}$ are.
> Our argument in the paragraph about the conditions for NPs is that since we deal with a target input $\tau\_{l-k+1:l}$ and target label $\tau\_l$ that can be thought of as sets with one element, Condition 3.1 is inherently satisfied.
> Hence, we only need to satisfy Condition 3.2 by redesigning the model architecture, which we do by average-pooling context features $r\_1, r\_2,\cdots r\_{l-1}$.
>
> As suggested, however, we added the following to the end of the 1st paragraph of the Page 4 to clarify the connection between $\mathcal{X}$ and $\mathcal{Y}$ notations and $\tau\_i$ notations,
>
> When a set contains only one element, its permutation is always itself.
> More formally, the consistency under permutation of Condition 3.1 in TPPs: $p\_\theta (\tau\_{l+1} \\, | \\, \tau\_{l-k+1:l}, \mathcal{C}\_l) = p\_\theta (\pi(\tau\_{l+1}) \\, | \\, \pi(\tau\_{l-k+1:l}), \mathcal{C}\_l)$, is satisfied since $\pi(\\{ \tau\_{l+1} \\}) = \\{ \tau\_{l+1} \\}$ and $\pi(\\{ \tau\_{l-k+1:l} \\}) = \\{ \tau\_{l-k+1:l} \\}$.
> Also, the marginalization under permutation in Condition 3.1 is satisfied as marginalization is not applicable for $p\_\theta (\tau\_{l+1} \\, | \\, \tau\_{l-k+1:l}, \mathcal{C}_l)$ since the target label set $\tau\_{l+1}$ contains only one element.
>
>
>
>
> > 3. How are the context features $r$ defined for TPP models?
>
> **Answer:**
> Please refer to the Q1 of the common questions above.
>
>
> > 4. Most notation in figure 1 $(r, G)$ is only introduced a few pages later in the paper, making it hard to interpret the figure the first time it's encountered.
>
> **Answer:** We understand the inconvenience.
> The global feature $G$ is the average-pooled feature of all the context features $r\_1, r\_2, \cdots, r\_{l-1}$.
> As suggested, we added the detailed description of the global feature right after they are mentioned along with the answer to the Q3, in the 2nd paragraph of the Page 4.
>
>
> > 5. What is the interpretation of the global feature $G$ and the latent variable z in the context of TPP modeling (section 3.2)?
>
> **Answer:**
> We agree that it was not stated clearly in the paper.
> We explained this point in detail in the Q2 of the common questions above, and reflected in the Appendix B. Please refer to them.
>
>
> > 6. Experiments in table 1 include accuracy scores (probably, for mark prediction?), but the rest of the paper only discusses unmarked TPPs.
>
> **Answer:**
> You are correct about the accuracy scores in Table 1.
> We did not add explanations about the extension to marked cases because it is simply extended by adding one or two layers in TPP literature.
> But, as suggested, we added the detailed description to the Q3 of the common questions, and Appendix G.

---

> ### Author Response · Authors · 2022-11-19
> **A Reminder to Reviewer Jm5S**
>
> Dear Reviewer Jm5S,
>
> As a reminder, the discussion session will end in few hours.
> We wanted to let you know that we have addressed all your questions and suggestions including unclarity of writing about some notations and model architecture as well as computation of evaluation metrics in inference.
> Furthermore, we added a detailed explanation for your great question about the interpretation of the global (latent) features.
> It would be highly appreciated if you can give us feedback about the points that are still not clear in our responses and the revision.
> We are willing to answer any doubts or questions until the last minutes.
> Thank you.

---

### Author Response · Authors · 2022-11-13
**Response to the Common Questions (3/3)**

> Q5. What is the structure of the decoder and what are the inputs and outpus of the decoder? -- Reviewer AxMo (Clarity Q2), dU5L (Weakness 1)

**Answer:**
As we specified in Section 3.3, the decoder takes the concatenated feature of $z, r\_l$, and $r^{\prime}$ for Attentive TPP where $z$ denotes one sample from either $p\_\theta(z \\, | \\, \mathcal{C}\_l)$ or $p\_\theta(z \\, | \\, \mathcal{C}\_L)$ (depending on training or inference time),
$r\_l$ denotes the feature of the target input $\tau\_{l-k:l-1}$, and $r^{\prime}$ denotes the feature from the attention module.
For the Meta TPP (without the attention module), the decoder takes as input the concatenated feature of $z$ and $r\_l$.
Here, $z, r\_l$, and $r^{\prime}$ are all $D$-dimensional vectors.

The decoder consists of two fully connected layers, and the input and hidden dimension of the decoder layers are either $2D$ or $3D$ depending on whether we use the feature from the attention module $r^{\prime}$.

The decoder outputs the parameters of the probability distribution of the next event time or $p\_\theta(\tau\_{l+1} \\, | \\, \tau\_{l-k+1:l}, z\_m)$.
Inspired by the intensity-free TPP (Shchur et al., 2020), we use a mixture of log-normal distributions to model the probability distribution.
Formally, for $l \in [1, L-1]$, $\tau\_{l+1} \sim MixLogNorm(\mu\_{l+1}, \sigma\_{l+1}, \omega\_{l+1})$ where $\mu\_{l+1}$ are the mixture means, $\sigma\_{l+1}$ are the standard deviations, and $\omega\_{l+1}$ are the mixture weights.
As suggested by the reviewer, we added the above description to the end of Section 3.3.

---

### Author Response · Authors · 2022-11-13
**Response to the Common Questions (2/3)**

> Q3. How is the Meta TPP extended to the marked cases? -- Reviewer Jm5S (Weakness Q6), YAR9 (Comment Q1)

**Answer:**
We extended the proposed method to the marked cases by adding a class prediction branch following Shchur et al.
Suppose a mark at $l+1$-th event is denoted as $y\_{l+1}$.
For the proposed Meta TPP, we compute the log-likelihood of the mark as,
$$ \log p\_\theta (y\_{l+1} \\, | \\, \tau\_{l-k+1:l}, y\_{l-k+1:l}, \mathcal{C}\_l) = \log \int p\_\theta (y\_{l+1} \\, | \\, \tau\_{l-k+1:l}, y\_{l-k+1:l}, z) p\_\theta (z \\, | \\, \mathcal{C}\_l) dz. $$

$\mathcal{C}\_l$ includes both event times and corresponding labels.
For implementation, we added one fully connected layer that takes as input the same features for the decoder (that predicts the next event time), and outputs the logits for classification.
A class prediction is made by taking argmax over the probability which is approximated using MC samples as,
$$ p\_\theta (y\_{l+1} \\, | \\, \tau\_{l-k+1:l}, y\_{l-k+1:l}, \mathcal{C}\_l) \approx \frac{1}{M} \sum\_{m=1}^M p\_\theta (y\_{l+1} \\, | \\, r\_l, z\_m)$$
Note that inputs $\tau\_{l-k+1:l}$ and $y\_{l-k+1:l}$ are encoded to $r\_l$.
We added this to Appendix G.

> Q4. How are the evaluation metrics -- NLL, RMSE, and Acc, computed in inference? -- Reviewer Jm5S (Weakness Q8), YAR9 (Actionable Feedback Q1), AxMo (Weakness Q5)

**Answer:**
As suggested, we provide how evaluation metrics are computed in inference.
Unlike Equation (7) in the main paper where ELBO is computed using samples from $p\_\theta(z \\, | \\, \mathcal{C}\_L)$, in inference, we do not have access to $z \sim p\_\theta(z \\, | \\, \mathcal{C}\_L)$.
But, as $p\_\theta(z \\, | \\, \mathcal{C}\_l)$ is trained to be similar to $p\_\theta(z \\, | \\, \mathcal{C}\_L)$ through $KL(p\_\theta(z \\, | \\, \mathcal{C}\_L) \\;||\\; p\_\theta(z \\, | \\, \mathcal{C}\_l)$, we use samples $z \sim p\_\theta(z \\, | \\, \mathcal{C}\_l)$.
As specified in Appendix I, we use 256 samples to have good enough approximation.

* NLL -- We approximate a log-likelihood of the next event time $\tau\_{l+1}$ using Monte-Carlo approximation as,
\begin{align}
    \log p\_\theta(\tau\_{l+1} \\, | \\, \tau\_{l-k+1:l}, \mathcal{C}\_l)
    &= \log \int p\_\theta(\tau\_{l+1} \\, | \\, \tau\_{l-k+1:l}, z) p\_\theta(z \\, | \\, \mathcal{C}\_l) dz \\\\
    &\approx \log \frac{1}{M} \sum\_{m=1}^M p\_\theta (\tau\_{l+1} \\, | \\, \tau\_{l-k+1:l}, z\_m)
\end{align}
where $M$ is the number of samples from $p\_\theta(z \\, | \\, \mathcal{C}\_l)$.

* RMSE --
We use a mixture of log-normal distributions to model $p\_\theta(\tau\_{l+1} \\, | \\, \tau\_{l-k+1:l}, z)$.
Formally, for $l \in [1, L-1]$, $\tau\_{l+1} \sim MixLogNorm(\mu\_{l+1}(z), \sigma\_{l+1}(z), \omega\_{l+1}(z))$ where $\mu\_{l+1}(z)$ are the mixture means, $\sigma\_{l+1}(z)$ are the standard deviations, and $\omega\_{l+1}(z)$ are the mixture weights.
Note that the parameters are the outputs of the decoder given a latent sample $z$.
Knowing this, we can analytically compute the expected event time for a latent sample $z$ with $K$ mixture components as,
$$E\_{\tau\_{l+1} \sim p\_\theta (\tau\_{l+1} \\, | \\, \tau\_{l-k+1:l}, z)} [\tau\_{l+1}] = \sum\_{k=1}^K \omega\_{l+1,k}(z) \exp{(\mu\_{l+1,k}(z) + \frac{1}{2} \sigma\_{l+1, k}^2(z))}.$$
Note that since this expectation is over $p\_\theta (\tau\_{l+1} \\, | \\, \tau\_{l-k+1:l}, z)$ where $z$ is one sample from the posterior, we need to take another expectation over the posterior as follows,
\begin{align}
    &E\_{\tau\_{l+1} \sim p\_\theta (\tau\_{l+1} \\, | \\, \tau\_{l-k+1:l}, \mathcal{C}_l)} [ \tau\_{l+1} ]\\\\
    &= E\_{z \sim p\_\theta (z \\, | \\, \mathcal{C}\_l)}  E\_{\tau\_{l+1} \sim p\_\theta (\tau\_{l+1} \\, | \\, \tau\_{l-k+1:l}, z)} [ \tau\_{l+1} ] \\\\
    &= E\_{z \sim p\_\theta (z \\, | \\, \mathcal{C}\_l)} \sum\_{k=1}^K \omega\_{l+1,k}(z) \exp{(\mu\_{l+1,k}(z) + \frac{1}{2} \sigma\_{l+1, k}^2(z))} \\\\
    &\approx \frac{1}{M} \sum\_{m=1}^M \sum\_{k=1}^K \omega\_{l+1,k}(z\_m) \exp{(\mu\_{l+1,k}(z\_m) + \frac{1}{2} \sigma\_{l+1, k}^2(z\_m))}
\end{align}
where $M$ is the number of samples from $p\_\theta(z \\, | \\, \mathcal{C}\_l)$.

* Accuracy -- We obtain class predictions by taking argmax over the probability distribution of class labels as follows,
$$arg\\,max\_{c \in [1, C]} p\_\theta (y\_{l+1} \\, | \\, \tau\_{l-k+1:l}, y\_{l-k+1:l}, \mathcal{C}\_l)$$
where $C$ is the number of marks.
The probability distribution of class labels is approximated using MC samples as,
\begin{align}
    p\_\theta (y\_{l+1} \\, | \\, \tau\_{l-k+1:l}, y\_{l-k+1:l}, \mathcal{C}\_l) &= \int p\_\theta (y\_{l+1} \\, | \\, \tau\_{l-k+1:l}, y\_{l-k+1:l}, z) p\_\theta (z \\, | \\, \mathcal{C}\_l) dz \\\\
    &\approx \frac{1}{M} \sum\_{m=1}^M p\_\theta (y\_{l+1} \\, | \\, r\_l, z\_m)
\end{align}
where $M$ is the number of samples from $p\_\theta(z \\, | \\, \mathcal{C}\_l)$.

We added this to Appendix C to avoid the confusion raised by the reviewers.

---

### Author Response · Authors · 2022-11-13
**Response to the Common Questions (1/3)**

We thank all the reviewers for their constructive feedback.
We tried to clarify all the questions and concerns raised by the reviewers, and reflect them to the revised pdf in blue.
In this comment, we responded to the questions raised by multiple reviewers; there are 5 overlapping questions.
We responded other questions directly to individual reviewers.
Please check out the responses and let us know if the reviewers still have any questions or doubts.


> Q1. How are the context features $r$ computed in encoder for TPP models? -- Reviewer Jm5S (Weakness Q3), YAR9 (Actionable Feedback Q2)

**Answer:**
Each context feature $r\_1, r\_2, \cdots, r\_{l-1}$ represents a feature from a transformer encoder such as Transformer Hawkes Processes (THP), that encodes the corresponding local history of the context set $\mathcal{C}\_l$.
For instance, $r\_i$ contains information of $\tau\_{i-k+1:i}$.
To make $r\_i$ only encode the subset of previous event times $\tau\_{i-k+1:i}$ (instead of the whole previous event times $\tau\_{\leq i}$), we masked out events that are outside of the local history window using an attention mask as shown in Figure 1(b) and (c), which is different from a regular attention mask shown in Figure 1(a).
For the transformer encoder, we use 2-layers with hidden dimension being $64$.
As suggested, we added the above description in the 2nd paragraph of the Page 4.



> Q2. What is the interpretation of the global  (latent) feature $G$ and $z$ in the context of TPP modeling? How does it make meta TPP different from previous TPP methods? -- Reviewer Jm5S (Weakness Q5), AxMo(Clarity Q1)

**Answer:**
As Reviewer Jm5S mentioned in the summary, the existing TPP models treat all event sequences in a dataset as realization of the same process.
In this work, on the other hand, the proposed Meta TPP treats each sequence as a realization of a distinct stochastic process.
We achieved this through the global (latent) feature $G$ and $z$ that capture the global and task-specific characteristics.
Recall the latent variable $z$ is the stochastic version of the global feature $G$.
Here, we assume there is noise for the latent feature, and it helps to capture more robust features to noise, which is, in particular, beneficial for TPP as it is a stochastic process.

For $G$ or $z$ to be task-specific, it has to be distinct for different sequences but similar throughout different event times $l \in [1, L-1]$ within the same sequence.
It is natural for the global features to be distinct by sequence but we need further guidance to make the global feature shared across all the event times in a sequence.
In fact, due to the permutation invariance constraint implemented in average-pooling, the global feature $G$ cannot be very different at different event time: adding some addition context features $r\_i$ will not change $G$ much.

For the latent variable $z$, additional guidance is provided, which is clearer with the objective of variational inference.
Recall the objective of the variational inference in Equation (6) is defined as,
$$arg\\,max\_\theta E\_{z \sim p\_\theta(z \\, | \\, \mathcal{C}\_L)}  \log p\_\theta(\tau\_l \\, | \\, \tau\_{l-k:l-1}, z)  - KL(p\_\theta (z \\, | \\, \mathcal{C}\_L) \\;||\\; p\_\theta (z \\, | \\, \mathcal{C}\_l)).$$

Here, regardless of the index of the target $l$, it always minimizes the KL divergence between $p\_\theta (z \\, | \\, \mathcal{C}\_L)$ and $p\_\theta ( z \\, | \\, \mathcal{C}\_l)$ where $L$ is the length of a sequence.
So, ideally, the latent variable $z \sim p\_\theta (z \\, | \\, \mathcal{C}\_l)$ should capture the same information as $z \sim p\_\theta (z \\, | \\, \mathcal{C}\_L)$.
It implies regardless of the index of the target $l$, the latent variable $z$ asymptotically captures the global feature of the whole sequence, which is equivalent to $z \sim p\_\theta (z \\, | \\, \mathcal{C}\_L)$.
Hence, the resulting $p\_\theta (z \\, | \\, \mathcal{C}\_l)$ captures the global and task-specific patterns, which ideally is similar to $p\_\theta (z \\, | \\, \mathcal{C}\_L)$.
As a result, the global (latent) feature is guided to be distinct for different sequences but similar throughout different event time $l \in [1, L-1]$ within the same sequence.

Back to our original goal: treating each sequence as a realization of a distinct stochastic process, we use the global (latent) feature that is distinct by each sequence to provide a task-specific information which is shared regardless of different event time step $l$.
It is neither implicitly nor explicitly considered in the supervised learning case.
In supervised learning, each event time step at each sequence is treated equally from which patterns for only one stochastic process is learned.

We added the above description in the Appendix B to make our argument clear.

---

### Author Response · Authors · 2022-11-16
**Friendly Reminder for Discussion - Ends in Less Than 4 Days**

Dear reviewers,

We appreciate the suggestions and feedback made by the reviewers so far, which have helped us to improve the contributions of this submission, including writing and empirical results. We would be more than happy to address any further concerns and questions about the submission. Please let us know if there are any additional points that you would like us to address. We appreciate your time and effort you put in for the reviews.

---

### Author Response · Authors · 2022-11-18
**Discussion Ends in about a Day**

Dear AC and Reviewers,

As a reminder, the discussion session will end in about a day. We have responded to all the questions and doubts raised by the reviewers and reflected them in the revised version (written in blue). But, if there are any further concerns or doubts, we are more than happy to discuss them so please free feel to let us know. Thank you.

---

### Author Response · Authors · 2022-11-19
**A Reminder for Discussion**

Dear AC and Reviewers,

As a reminder, the discussion will end in about 12 hours.
As a summary of the rebuttal, many reviewers agreed that our approach is novel and the experiment results are strong.
The concerns and suggestions raised by the reviewers were mostly about writing such as lack of description for the model architecture; please refer to the common questions for details.
Taking the feedbacks into account, we have addressed them all in the responses and reflected them in the revision.
In addition, we have provided the results for suggested experiments in the responses as well as in the revision.

We understand reviewing is such a time consuming and tedious work, but it would be very helpful if the reviewers can provide some feedbacks about our responses; any comments and criticism about which parts are clarified and which parts still need clarification would be highly appreciated. We look forward to hearing from the reviewers.
Thank you.

---

### Decision · Program_Chairs · 2023-01-20

**Decision:**

Accept: poster

**Justification For Why Not Higher Score:**

n/a

**Justification For Why Not Lower Score:**

n/a

**Metareview: Summary, Strengths And Weaknesses:**

The paper proposes a novel idea of formulating the TPP problem in a meta learning framework. Inspired by the NP literature, the authors present a conditional meta TPP formulation, a latent path extension, and an Attentive TPP model. The proposed methods achieve new SOTA on four TPP standard benchmarks. The reviewer agrees that the paper is a solid contribution of the paper. While Reviewer dU5L did not actively participate the discussion, I think the authors' rebuttal did a great job addressing the reviewer's concern. I therefore recommend acceptance of this paper to ICLR.

**Note From Pc:**

if the above contains the word "oral" or "spotlight" please see: "oral" presentation means -> notable-top-5% and "spotlight" means -> notable-top-25%. As stated in our emails, we are disassociating presentation type from AC recommendations